# A soft thermal sensor for the continuous assessment of flow in vascular access

Yujun Deng[1,2,3], Hany M. Arafa [4,5], Tianyu Yang[4,6], Hassan Albadawi[7,8], Richard J. Fowl[9], Zefu Zhang[8], Viswajit Kandula[4,5,10], Ashvita Ramesh[4,5,11], Chase Correia[4,5], Yonggang Huang [3,12,13,22] ✉, Rahmi Oklu [7,8,22] ✉, John A. Rogers [4,5,10,12,14,15,16,17,18,22] ✉ & Andrea S. Carlini [4,19,20,21,22] ✉

Hemodialysis for chronic kidney disease (CKD) relies on vascular access (VA) devices, such as arteriovenous fistulas (AVF), grafts (AVG), or catheters, to maintain blood flow. Nonetheless, unpredictable progressive vascular stenosis due to neointimal formation or complete occlusion from acute thrombosis remains the primary cause of mature VA failure. Despite emergent surgical intervention efforts, the lack of a reliable early detection tool significantly reduces patient outcomes and survival rates. This study introduces a soft, wearable device that continuously monitors blood flow for early detection of VA failure. Using thermal anemometry, integrated sensors noninvasively measure flow changes in large vessels. Bench testing with AVF and AVG models shows agreement with finite element analysis (FEA) simulations, while human and preclinical swine trials demonstrate the device's sensitivity. Wireless adaptation could enable at-home monitoring, improving detection of VA-related complications and survival in CKD patients.

Hemodialysis represents the most prevalent treatment option for chronic kidney disease (CKD), constituting a 32 billion USD healthcare burden affecting an estimated 800 million patients worldwide[1–6]. The current practice involves single sessions at specialized clinics lasting 4-8 h, repeated 3–7 times weekly to maintain blood filtration and homeostasis[7–9]. Patients undergoing regular treatment require surgical intervention to create a vascular access (VA) in the forearm or upper arm (Fig. 1). This involves establishing an anastomosis (connection) between a selected artery and nearby vein to create an arteriovenous fistula (AVF), or implanting a synthetic tube between the artery and

[1]State Key Laboratory of Mechanical System and Vibration, School of Mechanical Engineering, Shanghai Jiao Tong University, Shanghai, China. [2]Shanghai Key Laboratory of Digital Manufacture for Thin-walled Structure, Shanghai Jiao Tong University, Shanghai, China. [3]Department of Mechanical Engineering, Northwestern University, Evanston, IL, USA. [4]Querrey Simpson Institute for Bioelectronics, Northwestern University, Evanston, IL, USA. [5]Department of Biomedical Engineering, Northwestern University, Evanston, IL, USA. [6]School for Engineering of Matter, Transport and Energy, Arizona State University, Tempe, AZ, USA. [7]Division of Vascular and Interventional Radiology, Mayo Clinic, Scottsdale, AZ, USA. [8]The Laboratory for Patient-Inspired Engineering, Mayo Clinic, Scottsdale, AZ, USA. [9]Chair Emeritus, Division of Vascular and Endovascular Surgery, Mayo Clinic, Phoenix, AZ, USA. [10]Feinberg Medical School, Northwestern University, Chicago, IL, USA. [11]Massachusetts General Hospital, Boston, MA, USA. [12]Department of Materials Science and Engineering, Northwestern University, Evanston, IL, USA. [13]Department of Civil and Environmental Engineering, Northwestern University, Evanston, IL, USA. [14]Simpson Querrey Institute, Northwestern University, Evanston, IL, USA. [15]Department of Chemistry, Northwestern University, Evanston, IL, USA. [16]Department of Neurological Surgery, Northwestern University, Evanston, IL, USA. [17]Department of Electrical and Computer Engineering, Northwestern University, Evanston, IL, USA. [18]Department of Computer Science, Northwestern University, Evanston, IL, USA. [19]Department of Chemistry and Biochemistry, University of California at Santa Barbara, Santa Barbara, CA, USA. [20]Interdisciplinary Program in Quantitative Biosciences, University of California at Santa Barbara, Santa Barbara, CA, USA. [21]Center for Polymers and Organic Solids, University of California at Santa Barbara, Santa Barbara, CA 93106, USA. [22]These authors contributed equally: Yonggang Huang, Rahmi Oklu, John A. Rogers, Andrea S. Carlini. ✉e-mail: y-huang@northwestern.edu; Oklu.Rahmi@mayo.edu; jrogers@northwestern.edu; acarlini@ucsb.edu

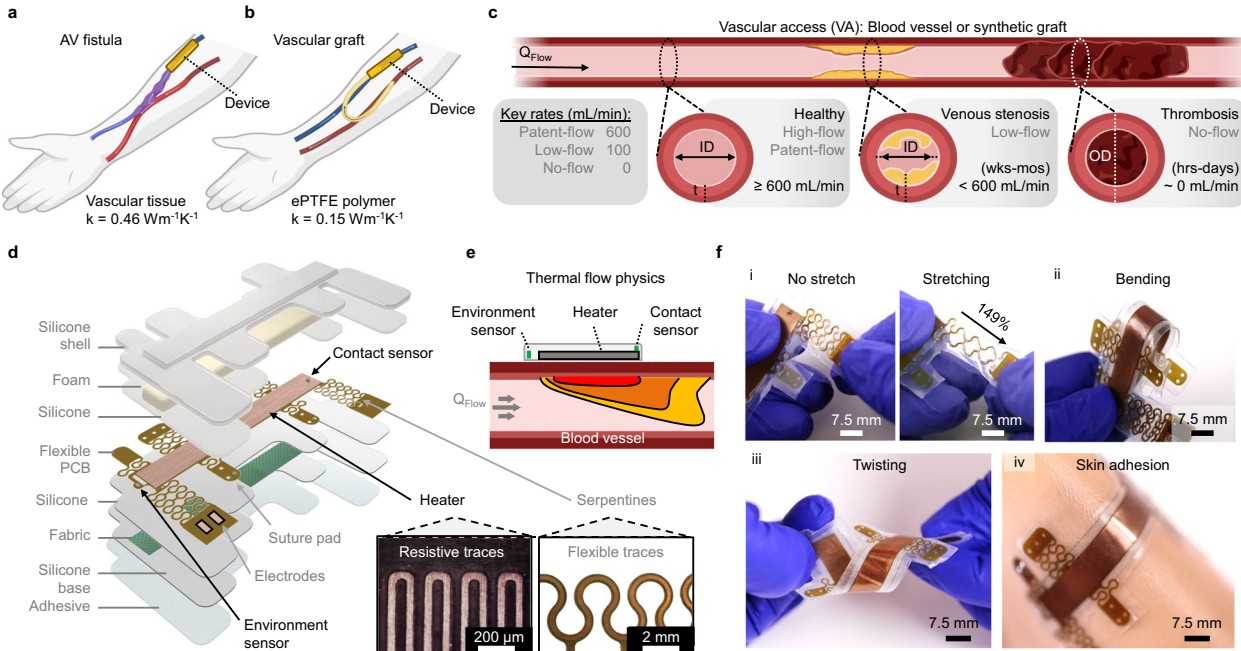

**Fig. 1 | Wearable thermal flow sensor for detection of vascular access (VA) stenosis. a, b** Diagram of VA fistula (**a**) and synthetic graft (**b**) with mounted devices. **c** Schematic of a healthy VA and VA failure by chronic venous stenosis or acute thrombosis. Key flow rate ($Q_{Flow}$) terminology is defined. **d** Exploded view of encapsulated device, containing a double-sided heater with thermal sensors to monitor device-tissue contact and environmental conditions. Magnified optical images show resistive heater traces and flexible suture pads for direct vascular application. Scale bars, 200 μm and 2 mm. **e** Schematic of thermal flow physics of the proposed anemometric sensing device. **f** Camera images of the flexible device for use on-skin and on-vessel. Demonstrations of (i) serpentine stretching, (ii) bending, (iii) twisting, and (iv) conformal adhesion to artificial skin. Scale bar, 7.5 mm.

vein to create an arteriovenous grafts (AVG) (Fig. 1a, b). This connection enables high arterial blood pressure to flow into the vein, ensuring adequate blood flow necessary for dialysis access. Clinically, healthy VAs require an inner diameter (ID) ≥ 6 mm, a tissue depth (*h*) of ≤6 mm (most are ≈1 mm), and a volumetric flow rate ($Q_{Flow}$) of ≥600 mL/min (Fig. 1c)[10,11]. Throughout this study, physiologically relevant healthy and unhealthy VA $Q_{Flow}$ is defined as patent-flow (600 mL/min), low-flow (100 mL/min), and no-flow (0 mL/min) (Fig. 1c inset, Supplementary Table 1). Specifically, our studies focus on detecting flow changes below these patent-flow conditions.

VA fistulas tend to last longer and are less prone to infection and clotting than synthetic grafts, but require months to mature fully for dialysis use[12]. Conversely, grafts can be employed for dialysis within days of implantation but necessitate long-term monitoring and frequent revision surgeries. Both approaches suffer from high unpredictable dysfunction and failure rates attributed to venous stenosis[13,14]. Venous stenosis commonly occurs in fistulas, characterized by vascular narrowing (stenosis) or complete blockage due to neointimal hyperplasia and fibrosis, which develops over weeks to months. Additionally, thrombosis, common with vascular grafts, results in acute occlusion within hours to days. Therefore, early detection of VA failure is crucial for timely interventions (e.g., thrombectomy) to prevent permanent failure. The lack of reliable, simple blood flow monitoring devices to prolong VA life in CKD patients undergoing hemodialysis represents an unmet healthcare need.

Direct assessment of VA flow represents the most useful indicator of graft/fistula patency. Therefore, deploying a reliable device for real-time blood flow readings within the VA would significantly enhance the standard of care for these patients. Standard modes of VA flow assessment include ultrasound, thermodilution, plethysmography, and blood pressure measurements (Supplementary Table 2). Previous approaches, ranging from implantable piezoelectric/capacitive sensors monitoring VA graft flow[15–17] to wearable optical sensors assessing VA patency, have shown mixed outcomes[18]. Implantable techniques

focus on the concept of a smart graft, exemplified by a recent study developing a 2 cm long piezoelectric sensor wrapping around the extraluminal surface of a PTFE graft[19]. However, ex vivo results do not fully capture the expected physiological flow within a VA graft. Another noninvasive wearable device (Graftworx) combines accelerometry and multichannel photoplethysmography with a proprietary algorithm to continuously assess graft patency, hemoglobin, and hematocrit levels in a VA[16,18]; however, this device fails to quantify $Q_{Flow}$ below 1000 mL/min. A highly sensitive, noninvasive vascular flow sensor capable of detecting changes in VA patency across physiologically relevant flow regimes will establish a new paradigm in the standard of care for dialysis patients.

Recent work with noninvasive calorimetric flow sensors present a promising alternative to previous technologies[20–25]. The development of an epidermal calorimetry-based sensor for ultralow flow conditions (0.01–1 mL/min) in hydrocephalus patients with ventriculoperitoneal shunts marks the first clinical deployment of thermal sensors[21,22]. However, calorimetry is unsuitable for the high flow environments inherent in healthy VAs given their diminished sensitivity at high $Q_{Flow}$, narrow detection ranges, and ambiguity to flow variations (non-monotonic response). Moreover, they require precise alignment of the device with its flow conduit (e.g., vessel). In contrast, anemometric sensors offer a simplified construction (integrated heater and sensor) and can tolerate minor user-induced misalignment[26]. An ideal device for detecting VA stenosis should demonstrate: (1) high precision, (2) a physiologically relevant response time in minutes, and (3) sensitivity to physiologically relevant flow. To meet these flow sensing requirements, we present a wearable, noninvasive, thermal anemometric flow sensing device for monitoring VA patency. A modular device designed for both skin and vessel measurements, along with testing in a large animal model, serves as a proof-of-concept study. Geometric optimization enables the device to interrogate a wide range of $Q_{Flow}$ (0–800 mL/min) in benchtop models of VAF and VAG. We further increase sensitivity to achieve instantaneous feedback in the presence

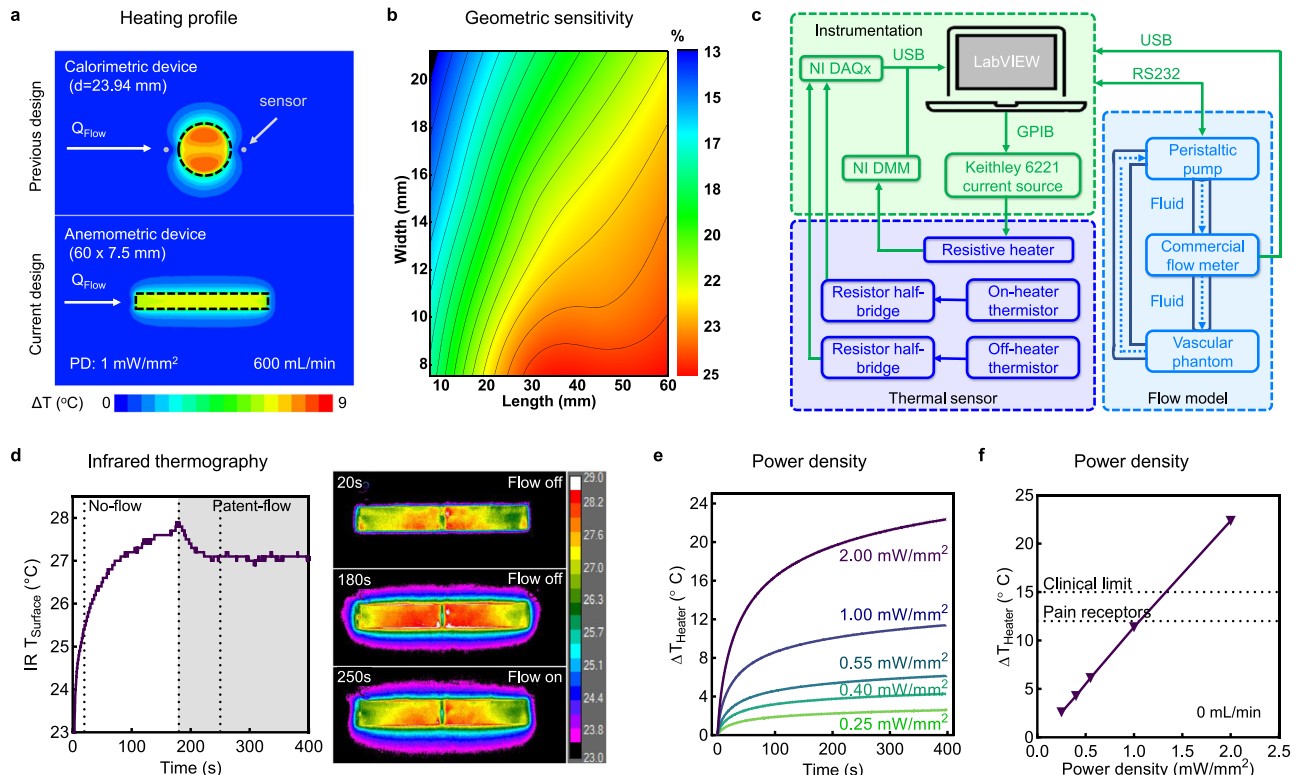

**Fig. 2 | Design and characterization of wearable flow sensing devices. a** FEA simulations of temperature profiles comparing an epidermally mounted thermal (top, previous design) and anemometric (bottom, current design) based sensor, with fixed heater areas of 450 mm². Simulations and experiments are performed under patent-flow conditions at heater PD of 1 mW/mm². **b** FEA simulations of heater sensitivity with respect to heater length and width modulation. **c** Block diagram of instrumentation, thermal flow sensor, and benchtop vascular flow model in experiments. **d** IR thermography of heater actuation under no-flow and patent-flow conditions. Selected images show net cooling at 250 s (patent-flow) as compared to 180 s (no-flow). **e** Time course analysis of heater temperature under no-flow conditions at 0.25, 0.4, 0.55, 1, and 2 mW/mm² PD ($n = 3$ technical replicates). **f** Corresponding steady-state heater $\Delta T$ after 400 s as a function of PD. Dotted lines indicate pain receptor activation and clinical dosing limits. Data are presented as mean values ± SEM. Source data are provided as a Source Data file.

of challenging physiological changes. Robust in vivo studies demonstrate accurate flow detection in real-time across various flow scenarios, including catheter-mediated injection of physiological solutions with variable temperatures, creating vascular hemorrhage followed by hemostasis, inducing stepwise vascular stenosis and reperfusion events, and real-time onset of thrombosis.

## Results

### Wearable sensor for the detection of flow in vascular access

Figure 1d displays a detailed diagram of the VA flow sensing device design. The flexible printed circuit board (f-PCB) comprises a double-sided Cu heater with an internal polyimide (PI) source, two negative temperature coefficient (NTC) sensors, and modular suture pads that connect via flexible PI serpentines for direct vascular application. Encapsulation of the electronics in EcoFlex-30, along with an ultrathin fiberglass fabric layer (25 μm), enhances flexibility and mechanical robustness, respectively. Devices adhere to the test tissue surface using a sacrificial silicone/acrylate skin-safe adhesive (180 μm). An insulating polyurethane (PU) foam layer (5 mm) and a minimal EcoFlex silicone layer (≈100 μm) are positioned above and below the f-PCB, respectively, to optimize and direct thermal conductivity (k) towards the tissue. Micro-CT 3D renderings depict the different layers of these encapsulated devices (Supplementary Fig. 1).

We measure high $Q_{Flow}$ using the principle of anemometry (Fig. 1e). This requires the use of a double-sided Cu heater as an actuator to heat the skin and as a temperature sensor to measure the temperature of the skin surface. Since blood in the subcutaneous vessels absorbs and transfers thermal energy, acting as a heat sink, there is a monotonic relationship between skin surface temperature

and blood flow, indicating that an increase in flow results in a decrease in skin temperature. Additionally, an environmental sensor is integrated to monitor local changes in passing blood temperatures, and a contact sensor is integrated on top of the heater, towards the device's edge, to assess conformal contact and potential delamination at the device-tissue interface. Specifically, poor contact results in a hotspot that the contact sensor can detect due to the lack of thermal transport from the heater into the bloodstream. Manipulation of encapsulated devices (Fig. 1f) without a foam layer demonstrates the device's flexibility and elasticity, allowing for conformal adhesion onto curved and soft surfaces.

### Design of anemometric flow sensing device and characterization

Finite element analysis (FEA) on a VA model enables optimizing and characterizing of the device's geometry and flow sensing mechanism (Fig. 2). Figure 2a illustrates a comparison of temperature profiles at the tissue surface for both a circular device (top) based on previous calorimetric designs[21,22], and the rectangular anemometric device (bottom) presented in this study. Both devices possess the same heater surface area (450 mm²), and are subjected to patent-flow conditions. The calorimetric device exhibits minimal thermal anisotropy between the upstream and downstream sensors, despite noticeable cooling along the blood flow path. Due to the non-monotonic response of calorimeters[27], lack of anisotropy could indicate high or no-flow conditions. Additionally, this mechanism heavily relies on accurate sensor alignment along the flow path, which can be difficult to see for patients with thicker skin. To overcome these challenges, we turn to anemometry which, offers a robust monotonic response to thermal dilution relative to $Q_{Flow}$. Flow sensitivity for an anemometric device heater or

contact sensor is defined as a function of temperature change ($\Delta T$) under low-flow and healthy flow conditions, using the following equation:

$$\text{Sensitivity} (\%) = \frac{\Delta T_{100\text{mL/min}} - \Delta T_{800\text{mL/min}}}{\Delta T_{800\text{mL/min}}} \times 100 \quad (1)$$

where $\Delta T_{100\text{ mL/min}}$ and $\Delta T_{800\text{ mL/min}}$ represent the steady-state $\Delta T$ after 400 s heating at 100 mL/min or 800 mL/min, respectively. A contour map of these sensitivities in Fig. 2b illustrates the impact of heater geometry on sensitivity, calculated using Eq. 1. Changing the shape from a circular to a rectangular heater and elongating it at a fixed surface area (450 mm²) increases this sensitivity (Supplementary Fig. 2), as more heat is directed above the blood vessel rather than to surrounding tissues. FEA simulations of various geometries at physiologically relevant VA $Q_{\text{Flow}}$ (0–800 mL/min) reveals the best thermal dissipation effect with a $60 \times 7.5$ mm² heater, the geometry employed for all experimental devices in this study. FEA temperature distributions on the skin surface and blood vessel wall after 400 s of heating further illustrates the advantage of this geometry (Supplementary Figs. 3 and 4).

Localized FEA measurements across the heater surface show the highest flow sensitivity at the furthest downstream location, which we chose as the location for our contact sensor (Supplementary Fig. 5). This component exhibits a stronger dependence on device-tissue contact than the heater, as indicated by thermal resistance calculations, allowing us to distinguish between good and poor conformal adhesion. Figure 2c and Supplementary Fig. 6 display the benchtop flow monitoring system, including (i) instrument control by a custom LabVIEW program for supplying direct current to the heater and monitoring temperature-dependent voltage and resistance values, (ii) analog front-end circuitry for the thermal sensing components, and (iii) a benchtop vascular flow model controlling $Q_{\text{Flow}}$ through a vascular phantom. We calculate a < 2% uncertainty in temperatures measurements, arising from power supply variations in these instruments (Supplementary Methods and Supplementary Table 3). Infrared (IR) thermography in Fig. 2d demonstrates the application of direct current and subsequent joule heating of a heater on a device conformally mounted atop a silicone phantom skin. A continuous resistive trace, replacing the dense array of surface-mounted resistors in previous systems, ensures spatially uniform heating and device pliability. Initiating patent-flow conditions at 180 s beneath the skin causes a 0.8 °C drop in surface IR temperature.

Device calibrations facilitate the conversion of measured resistance changes from the heater and sensors to temperature measurements (Supplementary Figs. 7–9). Heater resistance values are converted to temperature according to the positive temperature coefficient of resistance for copper (0.393% / °C). Recorded resistance values for each NTC sensor are related to temperature according to the extended Steinhart-Hart equation:

$$R = R_{\text{ref}}^{A + B/T + C/T^2 + D/T^3} \quad (2)$$

where $T$ is temperature, $R$ is measured resistance, $R_{\text{ref}}$ is the reference resistance at 25 °C, and A, B, C, and D are fitting coefficients (Supplementary Fig. 9). The standard protocol for each heating experiment involves 120 s of equilibration without heating, 400 s of heating, 400 s of no heating, and applying a low pass filter to all data (Supplementary Figs. 10 and 11), ensuring experimental precision and reproducibility. Figure 2e displays heater $\Delta T$ measurements upon application of direct current at variable power density (PD = 0.25, 0.4, 0.55, 1, and 2 mW/mm²). Steady-state temperature rises (defined as 400 s, unless otherwise stated) exhibit a linear dependence on PD under no-flow conditions (Fig. 2f). Subsequent experiments utilize a PD of 1 mW/mm², ensuring that maximal heating under no-flow conditions remain below the

threshold for high heat pain receptor activation (TRPV2 > 52 °C)[28,29] and within safe limits for clinical dermal devices[30]. Coalescence of heater and contact sensor temperatures under no-flow, along with reduced noise from air convection, confirms the integrated foam layer's efficient thermal insulation (Supplementary Fig. 12).

## Benchtop experiments with phantom skin and AVF models

Figure 3 analyzes the impacts of vascular flexibility, material dimensions, and skin on thermal flow sensing. Two phantom models, representing a VA fistula and graft, incorporate a high k ($\approx 0.4$ W·m⁻¹·K⁻¹) biomimetic vessel and a low k ($\approx 0.15$ W·m⁻¹·K⁻¹) medical grade ePTFE vascular graft, respectively (Fig. 3a and Supplementary Fig. 13). Both vessels are enveloped by a biomimetic adult human skin layer ($h = 1.89$ mm), atop which the device mounts for thermal flow sensing. Physical and thermal properties of each phantom model component are summarized in Supplementary Table 4. Flow sensitivities for both benchtop phantom models are presented in Fig. 3b and Supplementary Fig. 14. The relative flow sensing at steady-state ($\Delta T - \Delta T_{800\text{ mL/min}}$) exhibits a robust binary response in both phantoms between patent-flow and no-flow conditions. FEA simulations shows good agreement with these experimental values, with increasing sensitivity at disease-relevant flows (<400 mL/min) in the biomimetic vessels (Fig. 3c). Additionally, the first 20 s of heating help distinguish the impact of vessel wall conduction, aiding in the differentiation between a high k fistula and a low k graft (Supplementary Fig. 15).

Next, we explore changes in VA vessel $t$ and ID as a function of tissue remodeling (e.g., venous stenosis-induced wall thickening and stenosis). Measurements of both biomimetic and graft vessel compliance reveal bimodal responses to increasing $Q_{\text{Flow}}$, corresponding to an increase in biofluid pressure (Fig. 3d). The biomimetic vessel shows a significant monotonic increase in outer diameter (OD) and decrease in $t$, while the graft vessel exhibits limited compliance. FEA simulations of heater response under patent-flow (Fig. 3e) reveal negligible dependence on vessel OD but a strong dependence on $t$, as the vessel wall's conductive resistance ($R_{\text{cond}} = 2.5\ t \cdot \text{m}^2 \cdot \text{K} \cdot \text{W}^{-1}$) is more dimension-sensitive than the vessel diameter-related convective resistance ($R_{\text{conv}} = 0.15\ \text{ID} \cdot \text{m}^2 \cdot \text{K} \cdot \text{W}^{-1}$). Derivations of these values assume a fully developed velocity profile in the laminar regime (Re < 2300)[31]. Overall, higher flexibility, reduced $t$, and increased k of the biomimetic vessel over the graft significantly improve heat transport and flow sensitivity (24% vs. 1%, respectively, in Fig. 3c).

To understand the impact of transcutaneous measurements, we apply a skin layer ($h = 1.89$ mm) to both phantom vessels under binary patent-flow and no-flow conditions (Fig. 3f and Supplementary Fig. 16). Under no-flow conditions, the added skin layer acts as a heat sink (see black arrows). Slope analysis between no-flow and patent-flow conditions shows that thermal impedance from skin does reduce sensitivity. When this tissue depth is increased to 6 mm, representing the maximal tissue thickness for a patent VA, flow-dependent sensitivity drops from 24.2% (Fig. 3c) to 5.6% (Supplementary Fig. 17). Regardless, variations in tissue surface hydration do not impact our results, even after 9 thermal cycles (Supplementary Fig. 18). This is attributed to the large heater dimensions and prolonged actuation times that enable deep tissue measurements (>6 mm) and negate the impact of varying epidermal (~100 μm) thermal conductivities[27,32,33]. We further demonstrate reliable detection of transient $Q_{\text{Flow}}$ changes under thermal steady state (SS) conditions using multiple flow sweeps (cycling between 600 and 80 mL/min) (Fig. 3g).

## In vivo application of the VA flow sensing device

Large animal studies involving swine provide a close match to human vasculature and physiology[34,35]. Moreover, their femoral arteries possess equivalent geometries and blood flow dynamics to that of human fistulas (see Supplementary Table 5). The excess tissue depths of our swine femoral arteries (~20 mm), however, necessitate on-vessel

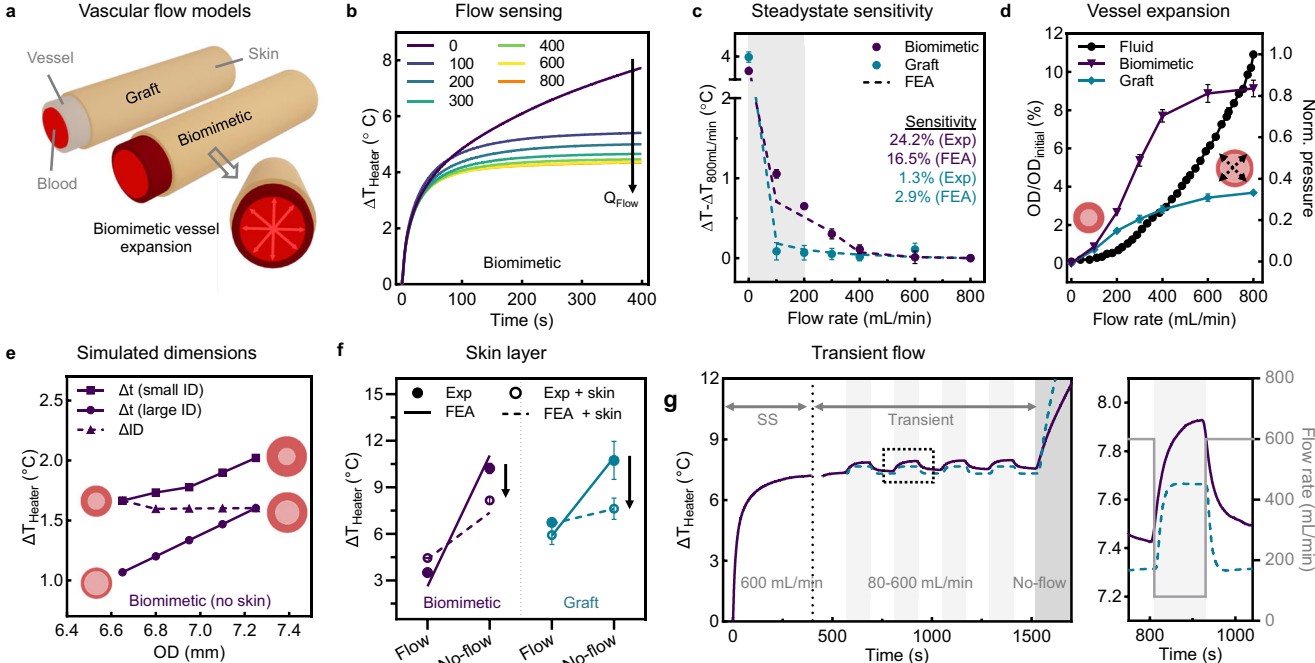

**Fig. 3 | Impact of vascular flexibility, material dimensions, and skin layer on flow sensing in artificial tissue models. a** Schematic of physiologically relevant benchtop vascular access phantom models of a synthetic graft and biomimetic fistula. **b** Representative flow sensing kinetics on the biomimetic model at $Q_{Flow}$ of 0, 100, 200, 300, 400, 600, and 800 mL/min ($n = 4$ technical replicates). **c** Relative flow sensing and calculated % sensitivity to $Q_{Flow}$ after steady-state heating for 400 s ($n = 4$ technical replicates). Sensitivity values are calculated according to Eq. 1. **d** Compliance in vessel-only biomimetic and graft models measured as the percent change in OD with respect to $Q_{Flow}$ and normalized inner fluid pressure ($n = 4$ technical replicates). **e** FEA simulations at patent-flow of physiologically relevant vessels with a fixed ID and variable $t$ for both a small (ID 5.95 mm, t

0.35–0.65 mm, square markers) and large vessel (ID 6.55 mm, $t$ 0.05-0.35 mm, circle markers). Dashed lines indicate vessels with fixed $t$ and variable IDs (ID 5.95-6.55 mm, $t$ 0.35 mm, triangle markers). **f** Impact of biomimetic skin layer ($h = 1.89$ mm) on steady-state temperature sensing for each phantom model under patent-flow and no-flow states. ($n = 4$ technical replicates). **g** Demonstration of transient response after reaching steady-state (SS) at patent-flow by modulation between patent-flow (600 mL/min) and low-flow (80 mL/min, shaded) conditions. Corresponding inset shows experimental (solid line) and FEA simulations (dashed line) of transient response to change in $Q_{Flow}$. All heating experiments performed at 1 mW/mm², and data are presented as mean values ± SEM. Source data are provided as a Source Data file.

interrogation for clinically relevant flow sensing (Fig. 4 and Supplementary Fig. 19). In vivo measurements of various tissues involve conformally affixing devices on the skin or a surgically exposed vessel (Fig. 4b). Control studies confirm that bending the device around an artificial vessel mimic and manual delamination from skin does not impact heater accuracy or cause signal loss, respectively (Supplementary Figs. 20–21). FEA simulations show good agreement with experimental measurements on different tissues (Fig. 4c), which are collected in tandem with Doppler ultrasound (Supplementary Fig. 22). Epidermal measurements (chest and neck) exhibit higher $\Delta T$ values in comparison with those on vessels (artery and vein), highlighting the predominant role of skin vs. blood k in heater measurements and significance of patent-flow in vessels for efficient thermal dissipation. Specifically, the device is effective in detecting flow through the carotid artery in the neck despite its substantial skin depth (20 mm by ultrasound) and heterogeneity (epidermal, dermal, subcutaneous fat, and muscle layers). This contrasts with chest measurements, which lack underlying blood flow. Furthermore, the lower time constants for vasculature measurements can enable shorter device operation times. Data from the environment sensor confirms identical incoming blood temperatures for vessels and rules out their contribution to differing contact sensor temperatures on the artery (Supplementary Fig. 23).

We perform quality checks of conformal contact for all in vivo and benchtop measurements. Specifically, an air gap at the device-artery interface ($k_{air} = 0.026$ W · m⁻¹ · K⁻¹) yields a significant increase in the contact sensor $\Delta T$ over that of the coupled heater (Δ+3.2 °C) (Supplementary Fig. 24). Conversely, we observed good agreement between the two measurements when applied to a vein. We suspect that device contact resistance ($R_{contact}$) is low (≈2000 mm² · K · W⁻¹) on the vein but high (≈6000 mm² · K · W⁻¹) on both tested arteries due to

arterial vasospasms. Arterial vasospasms during surgery are most apparent with Artery 1 (Fig. 4d), where reduced device-artery contact hinders heat transfer to the blood. This discrepancy between vasculature types is explained by the arteries' ability to contract in response to physical perturbations. Subsequent simulations verify this discrepancy when k is modulated at the device-vessel interface (Supplementary Fig. 25), showing excellent agreement with experimental results. For translational consideration, we note that dialysis through a fistula utilizes venous access and thus does not pose the same risk of vasospasms.

Next, we demonstrate that flow variations within a healthy human cephalic vein are detectable in real-time. An external load is applied at a distal position to an epidermally mounted device using a dynamic mechanical analyzer (Fig. 4e). After reaching steady-state tissue heating ($Q_{SS}$, 400 s), intervals of variable compression (10.0, 3.2, 2.7, 0.9, 0.4 N) reveal transient rises in heater temperature (Fig. 4f). Module displacement as a function of heater response ($\Delta\Delta T$) (Fig. 4g) reveals a region of partial venous occlusions (red shading) corresponding to a total displacement of 1.84 mm; this distance agrees with the ID of a female cephalic vein[36]. Response from the contact sensor confirms no perturbation of device-skin contact during these experiments (Supplementary Fig. 26). Occlusion at $Q_{SS}$ heating (90 s) shows a unique response from the environment sensor to local blood pooling proximal to the device as a transient drop in measured temperature (Supplementary Fig. 27 and Supplementary Movie 1). Conversely, the heater temperature rises at an increased rate due to reduced thermal dissipation. We note that mild temperature increases of the environment sensor (Δ+0.71 °C) match that of the skin surface temperature change (Δ+0.6 °C) during manual occlusion. This indicates a nominal influence of pooled blood and a dominating effect from limited

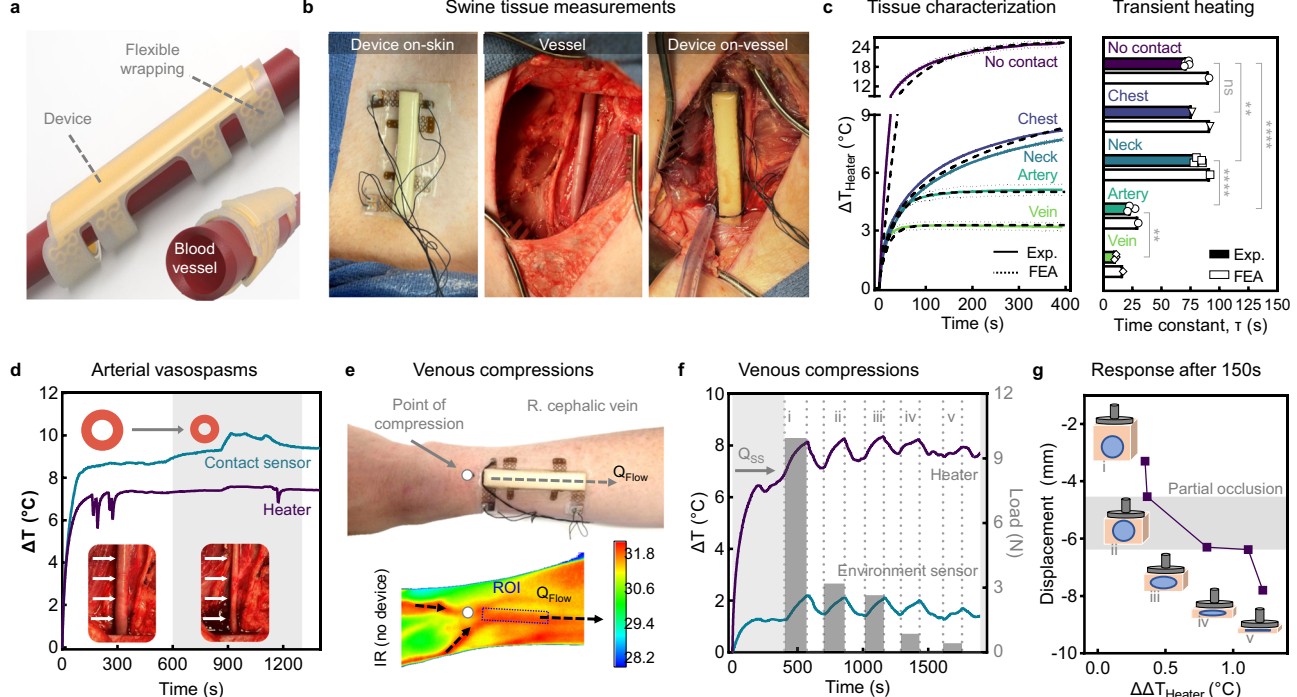

**Fig. 4 | In vivo device application and monitoring in swine. a** Image illustrating conformal wrapping of device around curved vasculature using flexible wrappings. **b** Camera images depicting an encapsulated device application on the skin (left), the exposed femoral artery in a swine before device application(middle), and after the device is mounted on the femoral artery (right). **c** Thermal measurements (left) and time constants (right) with no contact, chest, neck, artery, and vein tissues are depicted for experimental data and FEA simulations. Time constant $\tau$ is defined as the elapsed time when the heater transient $\Delta T$ reaches 63.2% of $Q_{SS}$ (no contact and neck $n = 4$ technical replicates, chest $n = 1$ technical replicates, vein and artery $n = 3$ technical replicates, collected for two pigs with the exception of chest measurements). **d** Device measurements prior to and during (shaded) vasospasms. Inset photograph shows the femoral artery before and during contraction. **e**–**g** On-body demonstration of varied blood flow using a device epidermally adhered over the cephalic vein. Camera image (**e**) of arm with mounted device and corresponding IR image of skin surface temperatures under the same treatment conditions without the device. The point of compression at 1 cm distal to device and direction of blood flow are indicated. Time-course thermal measurements (**f**) with the heater and environment sensor show heating to $Q_{SS}$ over 400 s, followed by manual occlusions and reperfusion events for 150 s each. Variable compressive loads are plotted on the right axis. Response reported as change in heater $\Delta T$ ($\Delta\Delta T$) after each 150 s of venous compression (**g**) with the region of partial occlusion indicated (shaded). Device operation at 1 mW/mm. Data are presented as mean values ± SEM. $^{ns}p = > 0.05$, $^{**}p \le 0.0025$, $^{****}p < 0.0001$. Ordinary one-way ANOVA with Šídák post hoc for pairwise comparisons. Source data are provided as a Source Data file.

convection. Reperfusion at 240 s causes a significant drop in temperature consistent with restored blood flow.

## Assessment of environmental fluctuation on flow sensing

Next, we investigate device sensitivity to rapid, local fluctuations in blood temperature and flow proximal to the device (Fig. 5). Sequential proximal injections of cold (−8 °C) and warm (+9 °C) saline boluses into the arterial lumen from an angiographic catheter tip positioned proximally to the measurement site show an immediate change in heater and sensor temperatures, followed by recovery of vessel surface temperatures (34 °C) to steady-state values (Fig. 5b, c). Exponential fitting of recovery profiles from the decoupled environment sensor (Fig. 5d and Supplementary Fig. 28) yields time constants ($\tau$) of 8.5 s and 12.8 s for warm and cold injections, respectively. Benchtop modeling of the biomimetic vessel and synthetic graft (Fig. 5e, f), with or without a skin layer, reveals similar trends for warm (+17 °C) and cold (−16 °C) saline injections. Notably, time constants for the biomimetic vessel ($\tau_{warm} = 8.1$ s and $\tau_{cold} = 11.3$ s) closely match those of the swine model (Fig. 5g). Thermal dissipation through skin delays response times approximately three-fold when the device is applied to skin. We observe similar trends with the low k graft model. Baseline drift is ruled out by measurements of ambient room, circulating biofluid, and saline temperatures (Supplementary Fig. 29). Furthermore, parallel testing with devices on control vessels (e.g., neighboring jugular vein and benchtop tributary vessels) reveal that adjacent blood flow effects do not impact local measurements (Supplementary Fig. 30). We next interrogate the device response to acute changes in

blood flow on a model for vascular hemorrhage and reperfusion as an advanced application (Fig. 5a and h). Steady-state heater $\Delta T$ increase from 5.98 °C (Step 1) under patent-flow conditions to 7.93 °C (Step 2) upon suture removal (39 s). This dramatic change is not detected with the environment sensor (Supplementary Fig. 31), showing that blood supply, and not its temperature, is altered during Step 2. Upon venous repair (Step 3, 310 s), heater $\Delta T$ drops rapidly when reestablishing blood flow. These studies underscore the sensitivity of our device to local changes in blood temperature under vascular injury conditions, regardless of vessel type or the presence of a skin barrier. This suggest our technology may be clinically useful for real-time detection of internal hemorrhages.

## Assessing the device's flow sensing capability in a swine model of endovascular occlusion

We percutaneously deliver a balloon catheter inside the common femoral artery to simulate significant hemodynamic changes during variable balloon inflation creating stenoses or complete occlusion (Fig. 6). Balloon inflation at proximal, medial, and distal positions relative to the device location (Fig. 6a, b) are visualized with digital subtraction angiography (Fig. 6c) and Doppler ultrasound imaging measurements (Supplementary Fig. 32). Device actuation during successive occlusion (balloon inflation) and reperfusion (balloon deflation) events (≈100–150 s each) (Fig. 6d) reveal position-dependent responses. At the proximal location, vascular balloon inflation triggers a rapid rise in heater and environment sensor $\Delta T$ values. Occlusion at the medial position causes a diminished signal from the environment

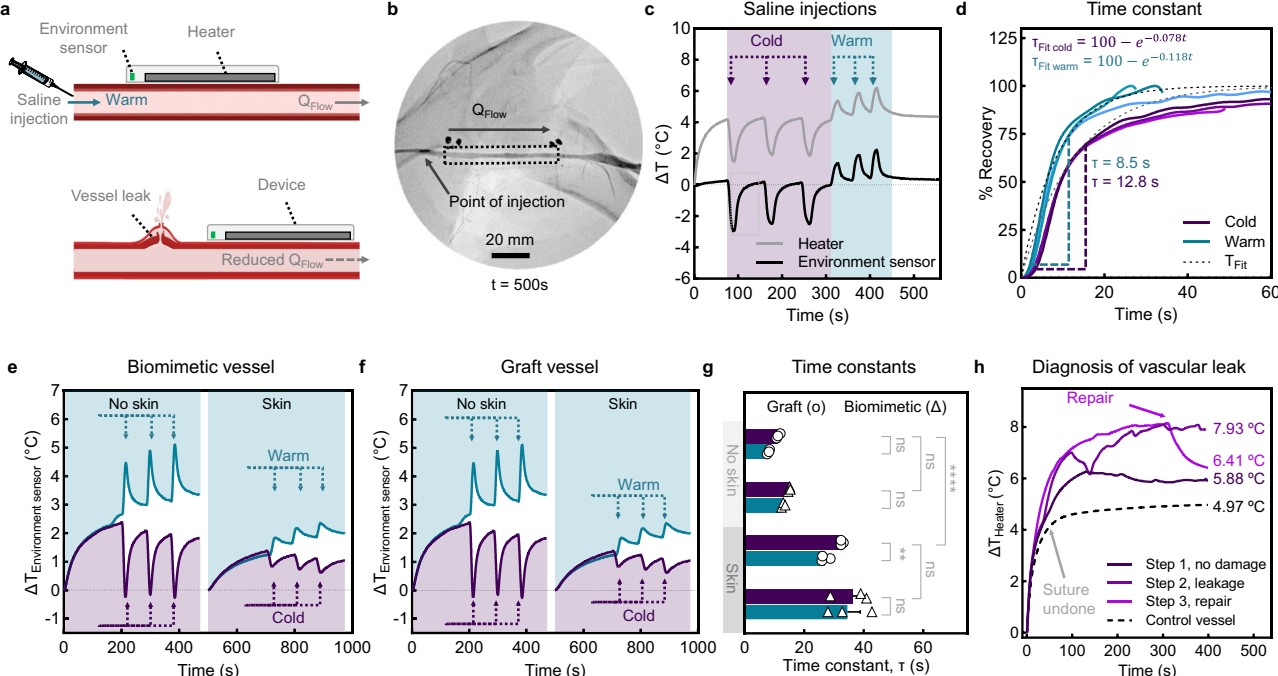

**Fig. 5 | In vivo assessment of local, rapid, temperature changes in the blood.**
**a** Schematics illustrating local perturbations to upstream blood temperature and supply, through saline injections and vascular tears, respectively. Injections with 30 mL of cold (−8 °C) and warm (+9 °C) saline in the artery occur proximal to the mounted device. **b** Angiogram of unobstructed artery with mounted device, point of injection, and direction of flow indicated. **c** In vivo flow sensing and rapid response to successive injections. **d** Percent recovery profiles for cold and warm saline injections fitted with an exponential form to yield time constants of 12.8 s and 8.5 s, respectively, to reach quasi-steady-state conditions defined as 63.2% of $Q_{SS}$. **e** Benchtop modeling of cold (−16 °C) or warm (+17 °C) saline injections through a biomimetic vessel at patent-flow without and with a biomimetic skin layer.

**f** Corresponding data with a synthetic graft. **g** Time constants for panels (**e**, **f**) ($n = 3$ technical replicates in triplicate per vascular models). **h** Time course data of venous exsanguination and repair of a swine (**e**). jugular vein showing differences in sensed $\Delta T$ under (step 1) no damage, (step 2) leakage following suture removal, and (step 3) repair through venous closure conditions. Simultaneous sensing of undamaged control vessel during step 2 shows no systemic response. Device PD is 1 mW/mm$^2$ for all runs. Data are presented as mean values ± SEM. $^{ns}p = > 0.05$, $^{**}p = 0.0072$, $^{****}p < 0.0001$. Two-way ANOVA with Šídák post hoc for pairwise comparisons of warm and cold injections into each phantom model. Tukey post hoc test is applied for comparisons between the graft and biomimetic vessel, with and without a skin layer. Source data are provided as a Source Data file.

sensor as unobstructed blood movement upstream cools the sensor. Repositioning to a medial from a slightly proximal position during the second cycle repeat better demonstrated this phenomenon. Distal occlusions are less apparent which we attribute to residual flow through collateral circulation. Minimal position adjustments reflect variable contributions of branched arteries to local perfusion, as shown by altered response profiles during each distal occlusion event. Overall, we observe a trend of decreasing time constants for occlusions ($\tau_{occlude}$) and heater $\Delta T$ as the site of stenosis transitions from a proximal to a distal position (Fig. 6e). Control measurements on an unobstructed neighboring vessel show no long-range dependence (Supplementary Fig. 33).

To validate the findings from the swine studies, we generate similar stenoses in benchtop biomimetic and graft vessels (Fig. 6f, g and Supplementary Fig. 34). It is important to note that these models do not reproduce positive pressure in human-like vasculature. In the absence of flow during proximal occlusions, the soft biomimetic vessel collapses and loses contact with the device. A bimodal response with the environment sensor exemplifies this due to progressive vessel deflation during proximal occlusions, followed by a data spike as the vessel re-inflates with flow (Supplementary Fig. 35). To maintain positive fluid pressure, only partial occlusions (75%) of the biomimetic vessel are performed in Fig. 6f. Synthetic graft vessels possess a higher modulus and thus do not experience this collapse. Overall, we observe similar trends to the in vivo data, with the response magnitude being highest and lowest for proximal and distal occlusions, respectively. Both in vivo swine and benchtop vessel data show a strong dependence on the location of vascular occlusion relative to time constants.

## Real-time detection of vascular occlusion and wearable device biocompatibility

A final demonstration of our device capabilities is the real-time detection of thrombosis, which is the most common and unpredictable mode of VA failure in patients receiving dialysis (Fig. 7). We induce an acute vascular occlusion within a segment of the femoral artery using a minimally invasive catheter-directed endovascular approach. Initial injections with thrombin are unsuccessful in generating a thrombus, likely due to anticoagulative heparin circulating in the bloodstream. Subsequent injection of gel embolic material (GEM, Obsidio™) vessel induces clot formation in the artery (Supplementary Fig. 36). Following equilibration (300 s), the device we detects injection, thrombosis, and reperfusion events (Fig. 7a). We note that heater signal accurately rebounds to pre-embolization flow temperatures ($\Delta T = 7.87$ °C) upon removal of the embolic material using a suction catheter. Reinjection of GEM generates a full thrombus and rapid increase in heater signal. Associated angiograms at 1680 s and 2400 s document the initial unobstructed (patent-flow) and later obstructed (no-flow) vascular site, respectively (Fig. 7b and Supplementary Movie 2–3).

Finally, we assess the biocompatibility of our thermal flow sensing devices. FDA safety guidelines allow an increase in local temperatures at the skin-device interface of up to $42 \pm 2$ °C, which corresponds to a 10 °C increase over normal skin temperature[37]. Our in vivo measurements show an increase of 3–8 °C during normal device operation on tissues with underlying blood flow, which does not exceed IEC clinical limit guidelines for devices in contact with tissue[30]. Furthermore, histological evaluation of harvested tissues show no remarkable change

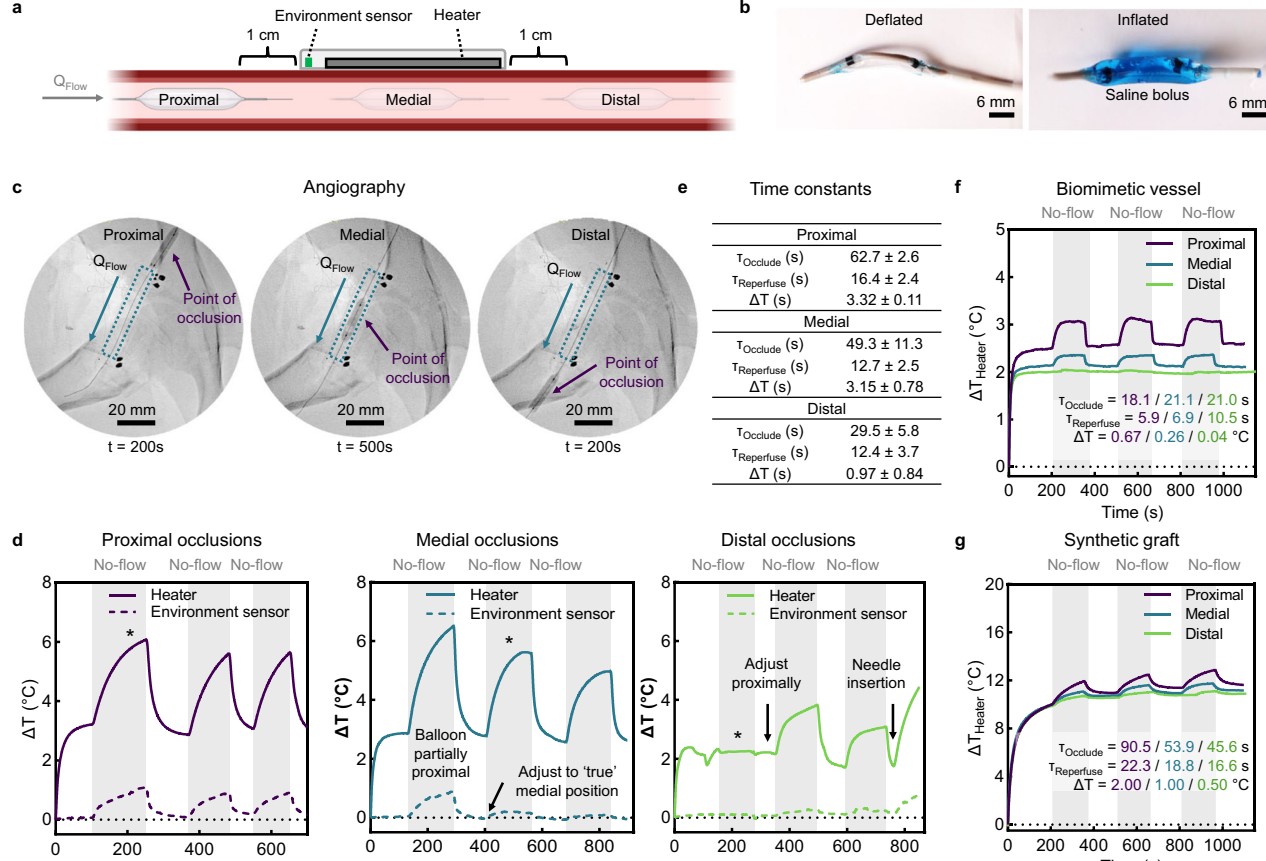

**Fig. 6 | In vivo assessment of flow-mediated by balloon catheter occlusions.**
**a** Schematic illustrating local occlusions at positions proximal, medial, and distal to the flow sensing device. **b** Images of deflated and inflated balloon catheter. **c–e** In vivo arterial occlusions and reperfusion events in a swine model. **c** Angiography of occlusions at locations proximal, medial, and distal to the mounted device. The point of occlusion and direction of flow are indicated with arrows. **d** Real-time flow sensing during successive occlusion (no-flow) and reperfusion (patent-flow) events at each location ($n = 1$ technical replicate measured in triplicate). Asterisks (*) indicate the time at which angiograms in (**c**) were recorded. **e**. Table summarizing

time constants for occlusion ($\tau_{\mathrm{occlude}}$), reperfusion ($\tau_{\mathrm{reperfuse}}$), and heater $\Delta T$ response to occlusion ($n = 3$ technical replicates, mean ± SD). **f, g** Real-time flow sensing during successive occlusions (75% for biomimetic vs 100% for graft) and reperfusion (patent-flow) events with the biomimetic vessel (**f**) and vascular graft (**g**) at each location ($n = 3$ technical replicates measured in triplicate). Inset data summarizes time constants for $\tau_{\mathrm{occlude}}$, $\tau_{\mathrm{reperfuse}}$, and heater $\Delta\Delta T$ after occlusion at each location. The PD is 1 mW/mm² for all experiments. Data are presented as mean values ± SEM. Source data are provided as a Source Data file.

in the vessel or skin morphology, despite continuous activation (Fig. 7c and Supplementary Figs. 37–38).

## Discussion

This study demonstrates a thermal flow sensing device that accurately monitors hemodynamic alterations in a vascular access observed in CKD patients receiving dialysis, including the adaptive response period during AVF maturation, vascular stenosis, acute thrombotic events, or complete failure. Compared with previous reports, our device demonstrates a monotonic response to flow changes with increasing sensitivity as a function of vascular stenosis. Through comprehensive flow-dependent benchtop modeling of both fistulas and grafts, we demonstrate how vessel material and geometry affect device sensitivity. We envision this platform to monitor progressive venous stenosis in a fistula or the rapid onset of thrombosis in a graft— both are primary clinical modes of obstructive VA failure. Ideally, a wearable thermal flow sensing device can detect problems at home or in a general clinical setting, such as during dialysis. Direct observation of temperature normalization during corrective measures establishes a new standard of care for dialysis patients in the clinic and at home. In theory, this sensitive feedback platform could find applications for the following users: (1) patients engaging in self-care to identify and manually extrude a forming clot without emergency surgery, (2)

surgeons fixing a tear or clearing a clot, and/or (3) healthcare workers tasked with preventing blood leakage during dialysis therapy.

## Limitations

As indicated by experimental measurements and simulation results, the temperature curves depend relatively weakly on flow rate in the regime of high flow rate. Thus, from a practical standpoint, experimental error due to environmental fluctuations or other forms of noise could be significant in such cases. One future approach to explore exploits co-integrated, adjacent flow sensors, to allow for differential measurements, with decreased sensitivity to noise. Another consideration in practical use is that continuous measurements for long periods of time may lead to adverse effects of cumulative heating. Because rapid changes in flow are not expected, in practice, the measurements will be performed in a low duty cycle mode, perhaps once every one or 2 h. In this way, cumulative heating can be neglected. Alternatively, the measurement accuracy can be improved by use of cooling rather than heating, simply because the threshold changes in temperature for activating pain receptors are larger for cooling.

Future iterations of this device will utilize lower PD and pulsed actuation to further reduce power consumption and total temperature change. Incorporation of closed-loop sensor feedback can prevent overactuation under no-flow conditions and enable transient actuation

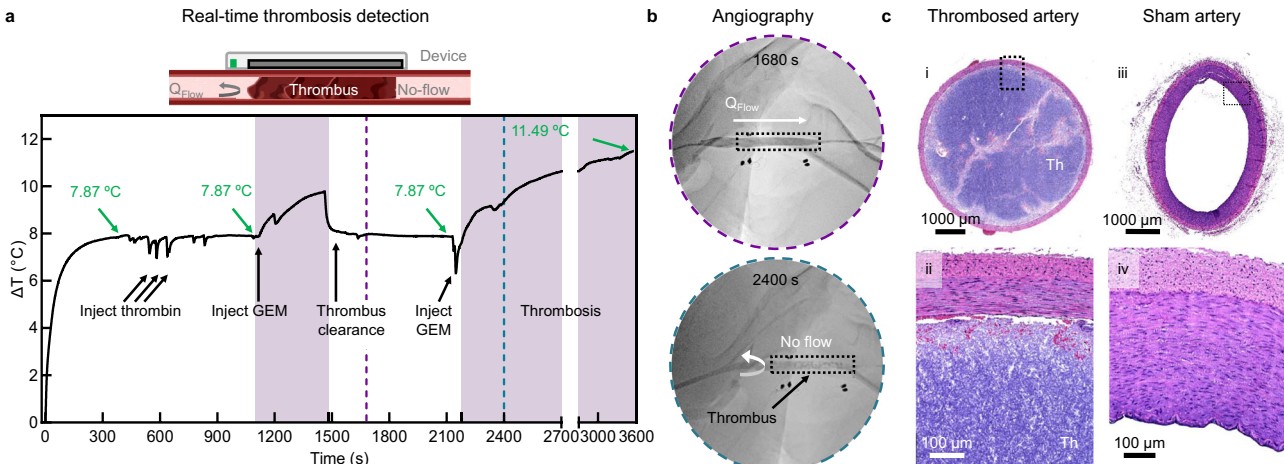

**Fig. 7 | In vivo detection of femoral artery occlusion in a swine model.**
**a** Schematic depiction of complete vascular occlusion due to thrombosis in a device-mounted blood vessel. Representative real-time thermal flow sensing before and after catheter mediated vascular occlusion using gel embolic material (GEM) to simulate thrombosis. (*n* = 2 animals). **b** Digital subtraction angiograms taken at 1680 s and 2400 s, depicting patent-flow at 1680 s and no-flow at 2400 s. The flow sensing device is depicted with a dashed outline and dark electrode pads, thrombus location is depicted with a black arrow, and the direction of flow is depicted with a white arrow. **c** Representative microscopy images of histology sections of thrombosed and sham control femoral artery at low (i, ii) and high magnifications (iii, iv) (*n* = 2 animals). Thrombosed blood is denoted as Th. The PD is 1 mW/mm². Scale bars, 100 or 1000 µm. Source data is provided as a Source Data file.

under flow. These devices will rely on information from contact and environment sensors to decide if physical adjustments of the adhered device or recalibration of baseline tissue temperatures is necessary, prior to making flow measurements. For wearable-only applications, where vessel mounting is not necessary, we can significantly reduce the form factor of our current device by removing pendant suture/electrode pads. Further miniaturization efforts will focus on increased flexibility of the device flexPCB by segmentation of the heater and anchoring sections with flexible serpentines to enable device adhesion over tortuous VAs. Shortening the overall heater length will also be advantageous in scenarios where superficial tissue depths vary along the vessel. We envision a wireless, wearable device that continuously measures VA patency exploiting low-cost commercial components (rechargeable Li ion battery, power management integrated circuits, analog front end, and active control circuitry for the actuator components of the circuit) and the integration of a Bluetooth low energy (BLE) system architecture to allow for seamless data transfer during normal user activities.

## Methods
### Fabrication of the thermal flow sensor
A laminate composite film of copper/polyimide/copper (18 µm/ 75 µm/18 µm, Pyralux AP8535R, DuPont Inc.) served as the substrate for the VA sensing device. An ultraviolet laser (Protolaser U4, LPKF) processed the film by ablating the copper layers, which patterned the traces, bond pads, and unplated vias. A structural copper divider (300 µm) was integrated within the center of the actuator to account for overlapping scan fields. Successive washes in stainless steel liquid flux (Worthington Inc), deionized water, and isopropanol (Fisher Scientific) prepared the resulting flexible printed circuit board (fPCB) for component population by removing remaining surface oxides. Thin flexible wire (36 AWG Copper Stranded Wire, Calmont Inc.) established an electrically conductive connection through via holes between the copper layers of the fPCB. A non-conductive epoxy (Loctite 3621, Henkel) mechanically bonded the surface mount components to the fPCB and reflow soldering with low-temperature solder paste (4900 P SAC305, MG Chemicals) established electrical contact between the surface mount thermistors (NTCG063JF103FT, TDK Corporation, Japan) and the copper pads to wired data acquisition electronics.

### Encapsulation and skin adhesive
(Poly)methyl methacrylate (PMMA) was spin-cast (3000 rpm, 1 min) and baked (180 °C, 3 min) to generate a solid hydrophobic film (≈700 nm) to facilitate encapsulated device removal by careful peeling. Encapsulation began by spin-casting (1500 rpm, 30 s) a thin layer (50 µm) of silicone prepolymer (1:1, Ecoflex 00-30, Smooth-On Inc., USA) onto the PMMA surface, gently adding a thin fiberglass fabric (25 µm) layer, allowing it to slowly settle and wick in the uncured silicone (30 s), and baking on a hotplate to cure (100 °C, 3 min). Another layer of silicone prepolymer (50 µm) was spin-casted (1500 rpm, 30 s), the fPCB was added and gently compressed into the silicone layer from above at room temperature (3 min) to remove air bubbles, followed by baking (100 °C, 3 min) to cure. Fresh silicone prepolymer was applied in excess to coat the device and placed in a vacuum desiccator (2 min) to fill and remove air gaps, spun (800 rpm, 30 s) to remove excess, and baked (100 °C, 3 min) to generate a defined device coating (100 µm). Devices at this step (340 µm thick) were used in Figs. 1 and 2 to demonstrate flexibility or for IR thermal characterization, respectively. Devices for benchtop and in vivo flow sensing were further insulated with a layer of polyurethane foam (DRX-3011, Dynarex CuraFoam, USA) that were laser cut into defined geometries (66 × 12 mm) and treated with waterproofing spray (275619, Rust-Oleum, USA). Briefly, a thin layer of flexible epoxy (Sil-Poxy, Smooth-On Inc., USA) was smeared onto the silicone surface, the foam layer was manually compressed on top of the encapsulated device, and allowed to cure (15 min) at room temperature. Fresh silicone prepolymer spin-coated (500 rpm, 30 s) and cured at room temperature (20 min) to generate an encapsulated, water-resistant topcoat (180 µm). Encapsulated working devices were peeled away from PMMA-coated slides and cut along the fPCB outline with ≈2 mm overlap. For on-tissue benchtop and in vivo testing, a CO₂ laser (Universal Laser Systems, USA) formed the outline of a commercially available medical-grade adhesive (3 M 2477 P) with a double-sided silicone-acrylate construction (180 µm), which was bonded to the bottom silicone of the encapsulated device. Peeling back the liner material on the skin-facing acrylate adhesive prior to tissue application allowed for conformal device contact.

### Micro-CT
MicroCT imaging was performed using SkyScan 1276 (Bruker, Kontich, Belgium) using an aluminum and copper filter, an X-ray source set at 90 kV, 200 µA current. Image stacks were acquired using 600 ms

exposure, at 20 μm pixel size, 360° rotation at 0.6° rotation steps, with two frames averaging. The 3D image stacks were reconstructed using NRecon software and InstaRecon CBR Server (version 1.7.4.6, Bruker, Kontich, Belgium) after adjustment for random movements, beam-hardening correction, ring artifact reduction, and smoothing. 3D rendering of the imaging stacks was visualized using CTVox software (version 3.3. 0 r1383, Bruker, Kontich, Belgium).

## Finite-element analysis

The commercial software Fluent (2020 R2 version) was used to investigate the thermal response in tissues caused by thermal actuation of the device, as previously reported[21,22]. Analyses were three-dimensional and transient, accounting for heat transfer in fluids and solids at ambient temperature. Two simulation models were constructed based on experimental setups discussed in this manuscript. The first model incorporated tissue, vessel, fluid, and device, with the device situated in a planar configuration above the tissue. The second model comprised only the vessel, fluid and device, with the device curved around the vessel directly. The fluid within the vessel was discretized using fine hexahedral elements, thereby ensuring the accuracy of the calculations. Power delivered to the heated area of the sensor matched experimental values (0.25, 0.4, 0.55, 1, and 2 mW/mm$^2$), and the temperature of the entire upper surface of the sensor was recorded (1 Hz). The outer walls were adiabatic except for the fluid inlet and outlet of each model. Initial temperatures of the fluid and solids were set to 300 K. The fluid inlet was maintained at a constant mass flow rate ($Q_{Flow}$) throughout the duration of a single simulation. Key thermal parameters used in the simulation included thermal conductivity, specific heat capacity, and density, as shown in Supplementary Table 4. Simulations of benchtop experiments defined water as the fluid, SynDaver skin as the tissue, and SynDaver vessel or ePTFE as the corresponding fistula or graft materials, respectively. Simulations of in vivo studies utilized human blood, skin, and vessel material parameters as model inputs. Geometric parameters such as vessel diameter, thickness, and depth within skin were varied for each simulation to match experimental conditions in this manuscript.

## Thermal transport analysis using a thermal equivalent circuit

A thermal resistance circuit (TRC) was built to analyze the thermal transport from the heater device through the vessel wall to the fluid flow in the vessel (Supplementary Figs. 5 and 24). Since the thermal boundary layer in the flow developed along the 60 mm long heater device, the heater was considered as three 20 mm long elements to depict different heat convection at the initial (Heater,0), medium (Heater,m), and sensor (sensor) parts of the heater. The three elements had identical heat input (Q) from the heater device. The majority of the input heat (Q) was transferred through the device, device-vessel interface, vessel wall and finally convected to the fluid flow. The rest of the heat ($Q_l$) was lost into the environment via natural air convection. Thermal impedances of each heat transfer barrier were calculated as thermal resistances in the table. The trace thermal resistance ($R_{trace}$) of the heater Cu resistive traces was calculated by $R_{trace} = (l_{trace} / k_{trace})/(A_{trace} / A_{element}) = 200000$ mm$^2 \cdot$ K/W, where $l_{trace} = 20$ mm was the element length, $k_{trace} = 20$ W/(m$\cdot$K) was the equivalent k of the heater, $A_{trace} = 0.75$ mm$^2$ was the heater cross-section area, and $A_{element} = 150$ mm$^2$ was the contact area between each element and the vessel. The device thermal resistance ($R_{device}$) from the heater to the device-vessel interface was $R_{device} = t_{device} / k_{device} = 2000$ mm$^2 \cdot$ K/W, where $t_{device} = 40$ μm was the device trace thickness and $k_{device} = 20$ W/(m$\cdot$K) was the trace equivalent k. The thermal resistance of the vessel wall was $R_{device} = t_{vessel} / k_{vessel}$, where $t_{vessel}$ was the vessel wall thickness and $k_{vessel}$ was the vessel thermal conductivity depending on the vessel dimension and material. The heat loss to environment $R_{loss} = 10000$ mm$^2 \cdot$ K/W was estimated from experiments. The flow convective resistance was calculated based on

the heat transfer coefficient correlation $h(Q_{Flow}, ID, x) = 5300k_{Fluid}/ID \cdot (Q_{Flow}/x)^{0.3172}$ in the thermal entry region of a hydraulically fully developed pipe flow[31] The thermal resistance of heat convection was $R_{conv} = 1/h_{average}$, where $h_{average} = \int_{x1}^{x2} i(Q_{Flow}, ID, x) dx / (x_1-x_2)$. Solving for the thermal equivalent circuit, the three elements' temperatures $T_{heater,0}$, $T_{heater,m}$, $T_{sensor}$ were obtained. The heater temperature was the average of the three elements temperatures, $T_{heater} = (T_{heater,0} + T_{heater,m} + T_{sensor})/3$. The sensor at the end of the heater had the temperature $T_{sensor}$. By matching the temperatures from the circuit analysis and the experiment, $R_{contact}$ between the device and vessel was obtained, ranging from 1000 mm$^2 \cdot$ K/W (good contact) and 20,000 mm$^2 \cdot$ K/W (poor contact) depending on the contact quality[38].

## Data acquisition and instrumentation

Simultaneous data acquisition for two devices occurs through the universal serial bus (USB) and general-purpose Interface Bus (GPIB) with a host laptop (Thinkpad T560, Lenovo). The entire system consists of two DC current sources (6220/2182 A, Tektronix Keithley), two 6.5-digit digital multimeters (DMM, USB-4065, National Instruments), and a multifunction I/O device (USB-6212 DAQx) (Supplementary Fig. 6). Acquired thermistor resistances are converted to a voltage using a potential divider (resistor half-bridge configuration) to create a ratiometric relationship between thermistor resistance and voltage output. All instrumentation and data were controlled and recorded via a custom interface (LabVIEW 2018, National Instruments), and processed with custom algorithms (MATLAB R2022b, Mathworks). Voltage samples are acquired at a sampling frequency of 5 Hz.

## Signal filtering and data analysis

Power spectral density calculations showing primarily low-frequency noise appear in Supplementary Fig. 11. A low pass filter (second order Butterworth infinite impulse filter) at 0.04 Hz was applied to the raw voltage signals to remove any high-frequency signatures. Zero-phase filtering removes any inherent phase lag and ambient 60 Hz noise related to thermal measurements. Data analysis of each device requires the conversion of measured resistances from two sensors and one actuator (Heater) into temperatures and then into $Q_{Flow}$. Unless otherwise stated, time constants ($\tau$) are calculated as the time it takes to reach quasi-steady-state conditions defined as 63.2% of $Q_{SS}$ values at a given $Q_{Flow}$.

## Heater and sensor calibration

The heater and sensor resistances were calibrated on a hotplate, surrounded by an air convection limiting barrier, to temperatures measured by IR imaging (A6255sc, FLIR Systems) and thermocouple readings (HH374, Omega Engineering Inc.). IR emissivity ($\varepsilon$) values for each device material were fitted using the known hotplate $\varepsilon = 0.91$ and thermocouple readings (Supplementary Figs. 7–8). Devices were adhered to a hotplate equilibrated at variable temperatures for 10 min, then 500 μA was supplied to the heater. Voltage measurements from the heater and sensors were collected with the DMM and DAQx, respectively, over 20 s intervals (5 Hz sampling) in triplicate for each device ($n = 6$ devices). Voltages were converted into resistances and plotted as a function of measured temperature. Fitting coefficients for the sensors were calculated using the Steinhart-Hart equation (Supplementary Fig. 9). Calibrations for the heater and sensors were compared with literature fits using the temperature coefficient of resistance (TCR) of copper ($3.9 \times 10^{-3}$/C) and manufacturer specifications for the thermistors, respectively.

## Benchtop vascular flow models

Flow through vascular phantom models was controlled with a peristaltic pump (M6-3L, U.S. Solid M6-3L) and monitored using an in-line turbine flow meter (FTB312, Omega) at representative blood $Q_{Flow}$ (0, 100, 200, 300, 400, 600, and 800 mL/min). A recirculating water bath

(2 L) maintained at 21 °C simulated artificial blood temperature. Vascular phantoms consisted of either a biomimetic vessel (SKU 131200, SynDaver) or medical grade ePTFE vascular graft (Gore-Tex®), with or without a surrounding layer of biomimetic skin (SKU 141500, SynDaver) (Supplementary Fig. 13 and Supplementary Table 4). Vessel expansion as a function of $Q_{Flow}$ was measured with a digital outside micrometer (IP65, Mitutoyo), and internal fluid pressure was measured with an in-line fluid pressure gauge at positions both immediately proximal and distal to the vessel of interest. Devices were conformally adhered on the vascular phantom surface along the path of flow, with the environment sensor situated proximally, under a fixed $Q_{Flow}$ (e.g., 100 mL/min). The standard thermal sensing protocol consisted of a no-heat thermal equilibration step (120 s), heating step (400 s), and no-heat cooling step (400 s) (Supplementary Fig. 10). Unless otherwise specified, heaters were actuated at 1 mW/mm².

### Benchtop saline injections and occlusions
Benchtop saline injections and balloon catheter occlusions utilized a branched tributary vessel for mounting a secondary control device for simultaneous measurements during local experiments on the primary vessel (Supplementary Fig. 29). The inlet for saline injections was situated 4 cm distal to the vascular branch and 4 cm proximal to the device location. A 4-channel thermoprobe (HH374, Omega) was used to monitor ambient room, circulating biofluid, vascular phantom surface, and injected saline (1x PBS) temperatures. Prior to injections, cold saline (4 °C) was stored in a freezer, and warm saline (37 °C) was incubated in a heated water bath. Syringe injections (30 mL over 10 s) of either cold or warm saline into the circulating biofluid (≈20 °C) were performed in triplicate at 85 s intervals (200, 285, 370 s device heating). Occlusions were simulated through inflation of a 4 mm × 15 mm Sceptor C occlusion balloon catheter (BC0415C, MicroVention Inc.) with saline containing blue food dye for visualization. To ensure full occlusion of test vessels, the deflated balloon was encapsulated in a thin layer (≈500–1000 μm) of EcoFlex 30. Occlusions for 150 s were performed in triplicate at positions 1 cm proximal, medial, or 1 cm distal from the device at 300 sec intervals (200, 500, 800 s device heating).

### In-vivo Swine model
Two male Yorkshire swine (S&S Farms, Brentwood, CA) weighing 52 ± 4.0 kg were premedicated using intramuscular injection of 5 mg kg-1 tiletamine-zolazepam (Telazol, Zoetis, NJ, USA), 2 mg ml-1 xylazine (Vedco inc., MO, USA), and 0.02 mg kg-1 glycopyrrolate (Wyeth, NJ, USA) followed by endotracheal intubation. Animals were placed in a supine position, and anesthesia was maintained using inhalation of 1–3% isoflurane at 1 L/min flow rate of 100 % O₂. Throughout the procedure, electrocardiogram, transcutaneous oxyhemoglobin saturation (SpO₂), end-tidal CO₂ concentration, inspired oxygen fraction, and core temperature were continuously monitored and documented. Ultrasound-guided (Butterfly iQ + , Butterfly Network Inc. Guilford, CT) common carotid access was performed to place a 5 French artery sheath (Cook Medical, IN, USA) followed by the introduction of a 5 French Cobra catheter (Cook Medical, IN, USA). The catheter was positioned proximal to the femoral artery, and digital subtraction angiography of the respective surgical side was performed at baseline using a floor-mounted mobile fluoroscope (OEC Elite C-Arm, GE HealthCare, Chicago, IL). Using ultrasound, the femoral artery and vein were identified and traced on the skin overlying the vessels. Over a 0.035-inch guidewire (Glidewire, Terumo, NJ, USA), a 6–8 millimeter mustang balloon dilatation catheter (Boston Scientific) was coaxially delivered to the femoral artery under fluoroscopic guidance. Standard Doppler ultrasound imaging of the femoral artery was performed with an ultrasound transducer (Butterfly iQ + , Butterfly Network Inc. Guilford, CT) at 8–10 MHz by applying the transducer directly onto the skin over the artery region to confirm arterial occlusion or patency with high specificity using B-mode, or color Doppler to measure

flow velocity. Digital subtraction angiography (DSA) at 8 frames per second acquisition was performed using a 10 mL injection of 1:1 dilution of iohexol contrast agent with physiologic saline (Omnipaque, 350 mg/ml; GE HealthCare); at ~5 mL/sec that was administered via the intra-arterial catheter. The thermal flow sensor was placed immediately anterior to the femoral artery over a length of 6 cm. The suture pads on the sensor (Fig. 1d) were wrapped around the femoral artery. The pads were sutured together with 4–0 silk suture allowing a non-constricting but snug fit around the artery. After all appropriate measurements were obtained, the animal was euthanized.

### Transdermal assessment of vascular flow
The common carotid artery and the jugular vein were traced on the skin using ultrasound then the flow sensor was placed over the skin over the marked locations using an adhesive film. The standard device operation protocol consisted of a no-heat thermal equilibration step (120 s), a heating step (400 s) for sensing, and a no-heat cooling step (400 s) (Supplementary Fig. 10). Unless otherwise specified, heaters were actuated at 1 mW/mm². Control flow measurements using ultrasound were correlated to the data obtained from the flow sensor.

### Subcutaneous assessment of vascular flow
To validate transdermal flow measurements and verify the effect of skin $h$ on the sensor accuracy, we placed the flow sensor directly over the vessel following surgical exposure. The femoral artery and veins bundle were identified and traced on ultrasound and by palpation, where the pulsation of the superficial part of the medial saphenous artery disappears in the skin fold between the gracilis and sartorius muscle. A 7 cm longitudinal superficial skin incision in the groin cranial to this point was made to avoid inadvertently damaging the medial saphenous vessels was made using electrocautery, and the subcutaneous tissues were dissected down using blunt-tip scissors. The fascia division of the sartorius and gracilis muscle was divided cranial to the penetration site of the medial saphenous vessels, first with small blunt-tip surgical forceps and then digitally. The two muscle groups were separated with a small self-retaining tissue retractor while taking care not to damage the femoral nerve and vessels. The artery and vein were exposed using blunt dissection, and the femoral vein was exposed just below and medial to the artery and femoral nerve. 3-4 drops of 2% lidocaine in isotonic solution were applied to the surface of the femoral artery and vein to prevent vasospasm. Scissors were used to enter the sheath and the femoral artery was dissected. A bolus of 150 IU/kg⁻¹ Heparin (Mylan, PA, USA) was administered intravenously to induce anticoagulation. Anticoagulation was titrated to reach an activated clotting time of over 200 s using an iSTAT system (Abbot Laboratories, IL, USA). After exposure, the sensor was then placed on top of the artery and fixed with sutures passed through soft tissue on both ends of the vessel. Blood flow distal to the flow sensor location was documented using ultrasound Doppler reading at baseline and after inducing partial or complete occlusion. The balloon inflation was adjusted to obtain Doppler values of 25%, 50%, 75%, or 100% correlating with data acquired from the flow sensor. This is important to confirm the sensitivity of the flow sensor reading using a clinically approved modality. Flow data acquisition was repeated 3 times with deflation of the balloon after each reading. Syringe injections (30 mL over 10 s) of either cold (25 °C) and warm (42 °C) saline through the catheter into the artery (surface temperature ≈34 °C) were performed in triplicate at 85 sec intervals (200, 285, 370 s device heating). Angiography of the vessel under patent-flow was used to visualize the catheter tip, positioned ≈1 cm proximal to the device location inside the swine femoral artery, before infusions. Thromboses were simulated by catheter mediated intra-arterial infusion of gel embolic material (GEM, Obsidio) at the device location. The standard device operation protocol consisted of a no-heat thermal equilibration step (120 s), heating step (400 s) for sensing, and no-heat cooling step

(400 s) (Supplementary Fig. 10). Unless otherwise specified, heaters were actuated at 1 mW/mm². At the conclusion of each experiment, the skin and the vessel segment that has the flow sensor application was explanted for histopathology.

### Forearm measurements and variable occlusions

On-body studies (IRB Protocol STU0020542, Northwestern Memorial Hospital, Chicago, IL) were carried out with informed consent from the participant. Devices were epidermally adhered over the right cephalic vein (visible through skin). IR thermography confirmed the path and location of this vessel. The acrylate side of the adhesive was mounted on the skin, and the device was allowed to equilibrate for three minutes. The standard device operation protocol consisted of a no-heat thermal equilibration step (120 s), followed by a continuous heating step at PD of 1 mW/mm². Manual occlusions and reperfusions of the cephalic vein were conducted by compression of the vessel through the skin with a plastic probe (1 cm²), followed by a release at 90 s and 150 s of heating. The point of compression (POC) was ≈1 cm proximal to the device. Identical experiments were conducted at the same location with the IR camera in the absence of the device to observe thermal changes in the skin surface as vasculature experienced blood flow disturbances. Automated occlusion and reperfusion events were simulated with a tensile tester (Mark-10, ESM303) equipped with a 100 N test module and 10 mm parallel plate geometry to apply variable force loads (10.0, 3.2, 2.7, 0.9, or 0.4 N for 150 s followed by 150 s at 0 N).

### Histopathology

Explanted tissues were fixed in 10% buffered formalin and processed for paraffin embedding using a standard protocol. Tissue sections were stained with H&E, as previously described[39]. To rule for vascular injury, morphometric evaluation was performed on serially cut sections by an operator blinded to the study.

### Statistics and reproducibility

No statistical method was used to predetermine sample size. No data were excluded from the analyses. The experiments were not randomized. There was no blinding in this study. This study employed six calibrated devices used interchangeably across each experiment without bias, and animal studies involved two swine.

### Reporting summary

Further information on research design is available in the Nature Portfolio Reporting Summary linked to this article.

## Data availability

All data supporting the findings of this study are available within the article and its supplementary files. Any additional requests for information can be directed to the corresponding authors. Source data and custom LabVIEW data acquisition files are provided with this paper. Source data are provided with this paper.

## Code availability

Relevant MATLAB scripts for data analysis and signal processing are uploaded to Code Ocean (DOI: 10.24433/CO.5920840.v1). Code for FEA calculations are available from Y.D.

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

## Acknowledgements

This work made use of the International Institute for Nanotechnology (IIN), the Keck Foundation and the State of Illinois, through the IIN. R.O. acknowledges support from NIH (R01CA257558, R01HL140951, R01DK130566, R01HL165176 and R01HL137193) and the Mayo Clinic Clinician Investigator Award. H.M.A acknowledges support from the National Institute of Neurological Disorders and Stroke of the National Institutes of Health under Award No. F31NS115422. Y.D. acknowledges support from the Natural Science Foundation of Shanghai (23ZR1428600). A.S.C thanks the University of California and a UCSB Faculty Research Grant for financial support. We thank the Querrey-Simpson Institute for Bioelectronics for support of this work. The authors are grateful to Natasha Cao for editing and proofreading assistance. The content is solely the responsibility of the authors and does not necessarily represent the official views of the National Institutes of Health. Figures 1c, 6a, and 7a were partly generated using Servier Medical Art, licensed under CC BY 4.0 (https://creativecommons.org/licenses/by/4.0/) and modified in PowerPoint. Figure 1a, b, and Supplementary Fig. 26a were created in BioRender. Carlini, A. (2024) https://BioRender.com/k53c078 and modified in PowerPoint.

## Author contributions

R.O. and J.A.R. were involved in the conception of the study. H.A., Y.H., R.O., J.A.R., and A.S.C. asked scientific questions and designed the experiments. V.K., A.R., and A.S.C. designed proprietary data acquisition protocols. A.S.C. fabricated devices, performed benchtop experiments, and analyzed the data. Y.D. and T.Y. performed computational simulations, and analyzed the data. H.A., R.J.F., Z.Z., and R.O. performed swine model surgeries and analyzed clinical imaging. H.M.A. and A.S.C. performed device measurements on swine models. C.C. generated 3D graphic renderings. A.S.C. wrote the manuscript. H.M.A., R.O., and J.A.R. acquired funding.

## Competing interests

The authors declare no competing interests.
