## [Transparent Peer Review file · Nature Communications]

A soft thermal sensor for the continuous assessment of flow in vascular access

Corresponding Author: Professor Andrea Carlini

Version 0:

Reviewer comments:

Reviewer #1

(Remarks to the Author)

In this manuscript, the authors report a skin-mounted blood flow sensor for detecting vascular access failure during hemodialysis. Their sensor can be used either on skin or directly on the blood vessel and is capable of measuring blood flow in both the high (up to 800 mL/min) and low (<0.01 mL/min) regimes, in contrast with previously reported sensors that are limited to high flow rates. They characterize the device in variety of experiments on benchtop and in a pig model, and notably demonstrate detection of induced stenosis when mounted on the skin. Overall, the manuscript takes a well-established sensor design and demonstrates improved blood flow monitoring capabilities in a convincing set of experiments.

My comments are below:

1. A confusing aspect of this manuscript is that results are reported for the device both on the skin (wearable) and on the vessel (implanted). Are the key performance metrics reported here valid for both modes of operation? If not, then clear differentiation is required (e.g. in a table). Given that the paper presumably focuses on wearable devices (title of the paper), why not move the vessel results into a supplemental figure and focus on the wearable case which is of more clinical interest?
2. It will be helpful to include a supplementary table summarizing previously reported approaches for VA flow measurement.
3. Can the authors comment on how variations the environment temperature and convection can be calibrated in their flow model using the sensor information?
4. The sensor requires significant instrumentation to be operated. Can the authors include discussion on future integration and miniaturization?
5. Fig. 3b,c. The curves for 600 and 800 mL/min seem difficult to distinguish. What is the criteria used to support the ability of the sensor to operate up to 800 mL/min?
6. Minor comments:
 - Line 55. Sentence fragment: "The procedure involves either."
 - Line 83. Reference needed.
 - Fig. 4g. Delta repeated.

(Remarks on code availability)

Reviewer #2

(Remarks to the Author)

The paper proposed a soft, wearable, flexible device designed to continuously monitor blood flow changes for the early detection of VA failure. By human and preclinical swine model testing, it has been proven that this device has certain potential for application in recording alterations in blood flow dynamics of target vessels. Although the aim of the study has

sort of application prospects, the research seems to be lack of innovation and sufficient content to support the conclusion. Some main issues are listed as follows.

1. From the experimental scene figure in the article, we find that the sensor needs to rely on external devices to record signals. Do these massive external devices prevent the sensor from truly being wearable?
2. Due to the fact that the sensor needs to be tightly attached to the surface of the skin during use, this is difficult to achieve in some cases, such as sweat on the skin surface. In addition, sensor performance should also be considered under different humidity and surface pH conditions to cope with potentially complex skin surface conditions.
3. This can be considered as a simple application of the existing measurement method based thermal anemometric flow sensing device, which has been well developed and frequently used. There was nothing new in terms of principles or models, its innovation does not meet the requirements of the journal.
4. The errors during the experimental process may come from various aspects, a clear and detailed analysis of uncertainty should be given.
5. The author uses a large number of letters and symbols in the text, and a nomenclature is necessary to help readers better understand the meaning of the symbols.
6. There are some grammar errors in the article. For instance, in Fig.3 caption, "FEA simulations (red dashed line) shown". It is necessary to double check the entire text carefully.

(Remarks on code availability)

Reviewer #3

(Remarks to the Author)

In this paper, the author describes the flexible and biocompatible thermal sensor for detecting vascular flow variation. The manufacturing process of the sensor was well-described, and in-vitro and in-vivo performance tests were done. I agree with the demand for and importance of this sensor for improving patients' QOL. However, I have several concerns in this paper. I request the author to answer the comments and revise the manuscript.

1. The development and evaluation process was organized very precisely. However, the final output of this paper was only detecting the flow rate variation and was insufficient as a scientific novel knowledge.
2. All of the figures on temperature variation were described using temperature difference Delta T. The absolute temperature value is also important in evaluating the harmfulness to the tissue. Therefore, please write the base temperature of Delta T to estimate the absolute value.
3. The author set the temperature to the operation limit of 52 degrees Celsius. It should be higher than the denaturation temperature of the protein. The author confirmed the thermal damage in the manuscript; however, the cumulative effect has not been considered. Please explain how the operation temperature was determined in this study.
4. From the viewpoint of detecting the flow rate variation, what is the minimum heater power to detect the temperature variation?
The author developed an accurate and fast response sensor on the device.
I think the author can reach the same conclusion even by reducing the heater power.
5. In an in vivo experiment, how much temperature fluctuates from biological activity?
In addition, what is the minimum heat power to generate a temperature variation more significant than that fluctuation?

(Remarks on code availability)

Version 1:

Reviewer comments:

Reviewer #1

(Remarks to the Author)

In this revision, the authors have included additional details that clarify the design of the sensor and the experiments. I have additional questions arising from the authors' response to the reviewers' comments.

1. There is a disconnect between what is claimed in the title/abstract and the reported sensor. As all reviewers have pointed out, the sensor's level of integration is below what is expected of paper published in this journal. While the authors make the point that the main advance is the optimized configuration of the heaters/sensors (and not the integrated system), this is not

clear from the title, abstract, and introduction. In my view, the integration of the sensor into a wireless device, while perhaps tedious, is of significant scientific value because it enables testing in continuous monitoring scenarios where a tethered setup cannot be used.

2. Based on the authors' response to my Q1, the current results do not demonstrate in vivo monitoring for BOTH the flow rate and vessel depth expected during vascular access monitoring. It remains unclear if the results from the porcine experiments are applicable when the sensor is separated by heterogenous tissue layers. Additional data should be included to support the effectiveness of the sensor, such as by placing artificial skin over the porcine vessel as suggested by the authors.

(Remarks on code availability)

Reviewer #2

(Remarks to the Author)

In this paper, a soft, wearable, flexible device was proposed to continuously monitor blood flow changes. The authors used in-vitro and in-vivo experiments to demonstrate that the sensor can sensitively detect vascular access failure during hemodialysis. It can be seen that the sensor has quite high sensitivity through the results presented in the paper. Although the sensor has the potential to be used in medical field in the future, there are still several unavoidable questions about the current work.

Some main issues are listed as follows:

1. In the paper, the authors have optimized the shape of the sensor, but whether the narrow and long sensor depends more on the precise alignment of blood vessels to ensure the accuracy of the measurement?
2. The author described the experimental content in great detail, in contrast, the analysis of the measurement principle and theoretical model is rough.
3. In the in-vitro experiment, the author used only one thickness of skin to attach to the blood vessels, and the experiment was inadequate and unconvincing.
4. Does the sensor need to be recalibrated each time the test object is changed?
5. In view of individual differences, whether differences in the skin thermal conductivity above the blood vessels affect the measurement results?
6. The network of blood vessels in the human body is very complex, whether the sensor has the same high sensitivity for tilted vessels with varying depth?

(Remarks on code availability)

Reviewer #3

(Remarks to the Author)

The author responded appropriately to the reviewer's comments.

(Remarks on code availability)

N/A

Version 2:

Reviewer comments:

Reviewer #1

(Remarks to the Author)

I thank the authors for their effort to address the questions raised. All of my concerns have been satisfactorily addressed. Regarding the title, I personally feel that "A soft thermal sensor for the continuous assessment of flow during vascular access" is both accurate and attractive. This avoids describing the sensor as both "wearable" and "wired", which is somewhat contradictory. I leave the final form at the discretion of the authors and editor.

(Remarks on code availability)

Reviewer #2

(Remarks to the Author)

In this paper, a soft, wearable, flexible device was proposed to continuously monitor blood flow changes. It can be seen that the authors did a lot of experiments to verify the performance of the sensor. The authors have explained some of the possible problems in the response letter. Beyond that, here are still some additional questions about the current work. Some main issues are listed as follows:

1. Given that we have done similar experiments before, it is generally easy to distinguish between fluid flow and unflow, but it is difficult to distinguish between fluids with varying flow rates. As can be seen in the article, the difference in temperature rise curves for different flow rates is very small. In fact, since the difference is so small that any experimental error may affect the sensitivity of the results, how can the authors accurately distinguish the temperature rise curves for different flow rates?
2. When the heating power is 1 mW/mm², the temperature rise is very close to that of the pain receptors, and while considering the measurement accuracy, the authors should consider reducing the heating power appropriately to alleviate the discomfort that may exist when wearing the product.
3. When wearing the measurement for a long period of time, is there any heat storage in the structure of the sensor itself, causing the sensitivity or accuracy of the measurement to be affected?

(Remarks on code availability)

Manuscript:

Wearable thermal anemometric sensor for the continuous assessment of flow in vascular access

REVIEWER COMMENTS

Reviewer #1 (Remarks to the Author):

In this manuscript, the authors report a skin-mounted blood flow sensor for detecting vascular access failure during hemodialysis. Their sensor can be used either on skin or directly on the blood vessel and is capable of measuring blood flow in both the high (up to 800 mL/min) and low (<0.01 mL/min) regimes, in contrast with previously reported sensors that are limited to high flow rates. They characterize the device in variety of experiments on benchtop and in a pig model, and notably demonstrate detection of induced stenosis when mounted on the skin. Overall, the manuscript takes a well-established sensor design and demonstrates improved blood flow monitoring capabilities in a convincing set of experiments.

My comments are below:

1. A confusing aspect of this manuscript is that results are reported for the device both on the skin (wearable) and on the vessel (implanted). Are the key performance metrics [AC1] reported here valid for both modes of operation? If not, then clear differentiation is required (e.g. in a table). Given that the paper presumably focuses on wearable devices (title of the paper), why not move the vessel results into a supplemental figure and focus on the wearable case which is of more clinical interest?

The primary purpose of this device is for on-skin measurements, and clinical studies with diseased patients bearing superficial VAs represent the best way to test our wearable system. For this initial proof-of-concept study, however, we utilized healthy subjects (humans and pigs). There were two key obstacles to achieving clinically relevant conditions:

1) We performed wearability measurements on a healthy human as a proof of concept for flow change detection below human skin (Fig. 4).

a. However, healthy human vasculature does not supply similar blood flow (e.g. cephalic vein is ~28 mL/min and 3.12 mm in diameter). Switching to a large animal model (e.g. porcine) allowed to us study physiologically relevant vasculature.

2) Porcine femoral arteries are a good match to fistula geometry and blood flow dynamics. Ideally, on-skin measurements would be best.

a. However, the significant mismatch in vessel depths between human fistulas (< 6 mm) and our swine model (20 mm) necessitated on-vessel measurements for more accurate testing conditions.

The main goal of positioning the flow sensing device directly on the vessel during the pig experiment was to demonstrate its sensitivity, regardless of its distance from the targeted vessel. This approach aimed to verify the consistency of flow readings obtained when the device is in direct contact with the isolated vessel compared to situations where the device is placed on the skin, further away from the vessel. This method helped eliminate potential interference from adjacent vasculatures and from subcutaneous soft tissues overlaying the vessel and ensure the reliability and specificity of the acquired flow measurements. Future studies with a porcine vascular access model will benefit from the use of an artificial human skin layer (SynDaver) over the vessel to improve our models.

Supplementary Table 5 and the following text has been provided to detail these differences:

“Large animal studies involving swine provide a close match to human vasculature anatomy and physiology^{1,2}. Moreover, their femoral arteries possess equivalent geometries and blood flow dynamics to that of human fistulas (see **Supplementary Table 5**). The greater tissue depths of our swine femoral arteries (~20 mm), however, necessitated on-vessel interrogation for clinically relevant flow sensing”....

2. It will be helpful to include a supplementary table summarizing previously reported approaches for VA flow measurement.

This is now included as **Supplementary Table 2**.

Supplementary Table 2. Previous approaches for VA flow measurements.

Approach	Advantages	Limitations	Reference
Ultrasound	 Wide flow range operation High flow rate sensitivity (> 180 cm/s) High accuracy (4% error) high penetration depth ($\geq 25\text{mm}$) 	 bulky form factor of handheld devices expensive requires skilled operator cause localized vascular compression subject to motion artifacts 	3,4
Thermodilution	 thorough hemodynamic evaluation, encompassing cardiac output and other relevant indices allows for calibration of pulse contour analysis, enabling continuous and real-time monitoring of cardiac output 	 requires large volume cold bolus injections (increased metabolic burden on patient) inconvenient for renal failure patients injection-based time-consuming only suitable for critically ill patients 	5
Plethysmography (PPG)	 measures arterial stiffness, pulse wave velocity, and blood pressure 	 low accuracy for flow rate measurements calculations of flow rate based on AI models from large datasets 	6
Blood pressure (in dialysis machine)	 assessment of cardiac function and arterial status monitored during dialysis sessions 	 does not calculate flow rates does not account for vascular dimensions, or blood viscosity only detects significant occlusions 	7

3. Can the authors comment on how variations the environment temperature and convection can be calibrated in their flow model using the sensor information?

Representative data on these measurements are shown in Figures 4d and 5. Supplementary Fig 12 shows the impact of thermal insulation above the heater and both sensors, thus eliminating the impact of significant air convection above the device. Supplementary Fig. 5 shows our threshold for “good” vs “poor” contact, in which poor contact is impacted by air convection between the device and tissue of interest. Supplementary Fig. 20 discusses the impact of manual device delamination and subsequent air convection.

As this study serves as a proof-of-concept analysis of sensor performance, we present both environment and contact sensor data separately to examine the impact of convection and incoming biofluid temperatures on our heater measurements, respectively. These direct measurements will inform on future iterations of a closed-loop device to accommodate fluctuations in the local environment when assessing temperatures measured by the heater. For instance, measurements can be paused if poor contact or significant changes to the environment sensor are detected, until corrected.

Please see response to question #4 for modifications to manuscript text.

4. The sensor requires significant instrumentation to be operated. Can the authors include discussion on future integration and miniaturization?

Response: We thank the reviewer for this comment. In the conclusion, we discuss future miniaturization and integration efforts. Below includes modified text for this section:

“Future iterations of this device will utilize lower *PD* and pulsed actuation to further reduce power consumption and total temperature elevation. Incorporation of closed-loop sensor feedback can prevent overheating under *no-flow* conditions and enable transient heating under flow. These devices will rely on contact and environment sensors information to decide if physical adjustments of the adhered device or recalibration of baseline tissue temperatures is necessary, prior to making flow measurements. For wearable-only applications, where vessel mounting is not necessary, we can significantly reduce the form factor of our current device by removing pendant suture/electrode pads. Further miniaturization efforts will focus on increased flexibility of the device flexPCB by segmentation of the heater and anchoring sections with flexible serpentine to enable device adhesion over tortuous VAs. We envision a wireless, wearable device which continuously measures VA patency exploiting low-cost commercial components (rechargeable Li ion battery, power management integrated circuits, analog front end, and active control circuitry for the actuator components of the circuit) and the integration of a Bluetooth low energy (BLE) system architecture to allow for seamless data transfer during normal user activities.”

5. Fig. 3b,c. The curves for 600 and 800 mL/min seem difficult to distinguish. What is the criteria used to support the ability of the sensor to operate up to 800 mL/min?

We agree that distinguishing flow rates differing between 600 and 800 mL/min is difficult (see Figure 3b). However, determination of flow changes within this range is not of clinical importance as both rates correspond to a healthy, or patent, VA (600-1500 mL/min). Our device shows increasing sensitivity to lower flow rates (<600 mL/min), which is of primary concern to clinicians when detecting VA failure. Flow rates below 600 mL/min represent abnormal conditions, and thus the important ranges to differentiate in this study. We chose to measure at 800 mL/min simply to confirm that sensitivity does not break down at higher flow rates. The manuscript text has been revised to remove the definition of *high-flow* (800 mL/min) and instead associate this rate with healthy conditions. The following text is also added:

“Although differentiation between healthy flow rates (600 and 800 mL/min) is difficult, sensitivity increases dramatically as a function of disease-relevant flow reduction in the biomimetic vessel (**Fig. 3c**).”

6. Minor comments:

- Line 55. Sentence fragment: "The procedure involves either."
- Line 83. Reference needed.
- Fig. 4g. Delta repeated.

These changes have been addressed in the manuscript.

Reviewer #2 (Remarks to the Author):

The paper proposed a soft, wearable, flexible device designed to continuously monitor blood flow changes for the early detection of VA failure. By human and preclinical swine model testing, it has been proven that this device has

certain potential for application in recording alterations in blood flow dynamics of target vessels. Although the aim of the study has sort of application prospects, the research seems to be lack of innovation and sufficient content to support the conclusion. Some main issues are listed as follows.

1. From the experimental scene figure in the article, we find that the sensor needs to rely on external devices to record signals. Do these massive external devices prevent the sensor from truly being wearable?

As this is a proof-of-concept study, we rely on a wired array to enable modular operation of the device (e.g. variable power density and measurement times of the heater), in addition to synchronous operation of multiple devices for control studies. It is common practice to prototype new devices with this external machinery. Conversion of this wearable wired device to a wireless construct is a rather straight-forward process and is now discussed as part of the next step in the conclusions section:

“We envision a wireless, wearable device which continuously measures VA patency exploiting low-cost commercial components (rechargeable Li ion battery, power management integrated circuits, analog front end, and active control circuitry for the actuator components of the circuit) and the integration of a Bluetooth low energy (BLE) system architecture to allow for seamless data transfer during normal user activities.

2. Due to the fact that the sensor needs to be tightly attached to the surface of the skin during use, this is difficult to achieve in some cases, such as sweat on the skin surface. In addition, sensor performance should also be considered under different humidity and surface pH conditions to cope with potentially complex skin surface conditions.

We expect that devices will be used by VA patients on dialysis, who are generally not physically active. In the event that humidity or sweat is a significant factor with conformal contact, there exist many commercially available thin film adhesives capable of waterproof and pH-insensitive adhesion (<https://www.science.org/doi/10.1126/sciadv.aau6356>) that can supplement the 3M 2477P adhesive (~180 um) used in this study.

3. This can be considered as a simple application of the existing measurement method based thermal anemometric flow sensing device, which has been well developed and frequently used. There was nothing new in terms of principles or models, its innovation does not meet the requirements of the journal.

Conventional thermal anemometric flow sensing devices are frequently used to measure the velocity of surface fluids in applications such as aircraft and fan blades. In this paper, we aim to measure the flow of blood vessels under the skin without direct contact with the fluid. As a result, the signal strength is weaker. Therefore, it is necessary to focus on theoretical research and explore wearable blood flow sensing. Although simple in application for direct measurements of homogenous materials (e.g. liquids, gasses), non-invasive measurements of heterogeneous materials with underlying fluid flow are much more nuanced. We believe this paper represents one of a very limited number of articles related to wearable thermal anemometry of biofluids:

Our device represents the first example of a wearable anemometer for such measurements of high flow rates (>100 mL/min) and large flow ranges (0-800 mL/min). For example, all previous wearable thermal sensors applied for hydrocephalus⁸⁻¹⁰ (calorimetry), sweat rate^{11,12} (calorimetry), and macrovasculature¹³ (calorimetry) cannot measure these conditions. In consideration of these unique requirements, our device architecture and assembly required a significant redesign within the bounds of a large parameter space (conversion from non-monotonic calorimetric sensing to anemometric sensing, actuator geometry and power density optimization, materials for integration and insulation, and flexible form factor).

This sensing architecture was specifically developed for this clinical application and the down-selection/optimization of certain parameters with the use of benchtop data and experimental design resulted in the bespoke design suited for the noninvasive measurement of VA patency. While there are other thermal anemometry-based devices which are used for a number of applications, there are no such devices which are designed and optimized specifically for this application.

Despite the significant need for at-home flow measurements of VAs, there are limited examples of applicable technology that can be used at-home and without skilled operators. We present the first iteration of such a device that has the capacity to be commercialized as a wireless sensor with an integrated mobile device app.

4. The errors during the experimental process may come from various aspects, a clear and detailed analysis of uncertainty should be given.

We agree that experimental error can arise from various sources and have added an uncertainty budget for the most critical components outlined in the manuscript. We now include the following text in our manuscript Main Text and Supplementary Methods section:

Main text: “We calculate a <2% uncertainty in temperatures measurements, arising from power supply variations in these instruments (**Supplementary Methods** and **Supplementary Table 3**).”

Supplementary Methods:

Uncertainty Analysis of Temperature Measurements.

To evaluate the accuracy and performance of temperature sensing circuitry implemented in this study, power supply variations are assessed. As shown in **Supplementary Fig. 6**, the half bridge configuration of NTCs (NTCG063JF103FT, TDK Corporation, Japan) used to measure changes in temperature includes a reference resistor and a 6.5 digit digital multimeter (DMM, USB-4065, National Instruments). **Equation 1** shows the relationship between the resolution of the analog to digital converter (ADC) within the DMM as a function of the output voltage and the supply voltage:

$$ADC\ output = \frac{V_{out}}{V_{SS}} * (2^N - 1) \quad (1)$$

Next, the temperature dependent signal (measurement across the thermistor) or the output voltage (V_{out}) can be seen in Equation 2:

$$V_{out} = \frac{R_{NTC}}{R_{NTC} + R_{REF}} * V_{dd} \quad (2)$$

Combining Equations 1 and 2 yields Equation 3:

$$ADC\ output = \frac{R_{NTC}}{R_{NTC} + R_{REF}} * \frac{V_{dd}}{V_{SS}} * (2^N - 1) \quad (3)$$

Since the reference voltage (V_{ss}) and the supply voltage (V_{DD}) are supplied from the same voltage source, they are effectively identical, which leads to Equation 4:

$$ADC\ output = \frac{R_{NTC}}{R_{NTC} + R_{REF}} * (2^N - 1) \quad (4)$$

Nominal values as well as expected errors for all of the components used in the temperature circuitry at included in **Supplementary Table 3**. Using Equation 2, the partial derivatives of error can be calculated. We find the simplified PDE and calculate that the maximum error contribution of this equipment is <2% at nominal temperature.

$$\sigma V_{out} = \sqrt{\left(\frac{\partial V_{out}}{\partial R_{NTC}} \sigma R_{NTC}\right)^2 + \left(\frac{\partial V_{out}}{\partial R_{REF}} \sigma R_{REF}\right)^2 + \left(\frac{\partial V_{out}}{\partial V_{dd}} \sigma V_{dd}\right)^2}$$

Supplementary Table 3: Nominal values and expected errors used in temperature circuitry.

Component	Parameter	Value	Unit
Thermistor nominal value (at 25 °C)	R_{NTC}	10k	Ω
Thermistor tolerance/error	σR_{NTC}	± 1	%
Reference resistor nominal value (at 25 °C)	R_{REF}	10k	Ω
Reference resistor tolerance/error	σR_{REF}	± 5	%
Supply voltage	V_{DD}	3.3	V
Supply RMS noise	σV_{DD}	1	mV

5. The author uses a large number of letters and symbols in the text, and a nomenclature is necessary to help readers better understand the meaning of the symbols.

We now include the following table as **Supplementary Table 1** in our manuscript.

Supplementary Table 1: Nomenclature and definitions.

Nomenclature		
Symbol	Unit	Description
ID	mm	Inner diameter of vessel
OD	mm	Outer diameter of vessel
t	mm	Thickness of vessel wall
h	mm	Depth of skin layer
Q_{Flow}	$\text{mL}\cdot\text{min}^{-1}$	Volumetric flow rate
k	$\text{W}\cdot\text{m}^{-1}\cdot\text{K}^{-1}$	Thermal conductivity
ΔT	$^{\circ}\text{C}$	Temperature change of heater or sensors from baseline values
$\Delta\Delta T$	$^{\circ}\text{C}$	Difference in temperature change during a measurement
PD	$\text{mW}\cdot\text{mm}^{-2}$	Power density of heater
R_{cond}	$\text{t}\cdot\text{m}^2\cdot\text{K}\cdot\text{W}^{-1}$	Conductive resistance of tissue
R_{conv}	$ID\cdot\text{m}^2\cdot\text{K}\cdot\text{W}^{-1}$	Convective resistance of tissue
Re	unitless	Reynolds number
$R_{contact}$	$\text{mm}^2\cdot\text{K}\cdot\text{W}^{-1}$	Contact resistance
Q_{SS}	$\text{mL}\cdot\text{min}^{-1}$	Flow rate at steady-state heating conditions
τ	s	Time constant, or time at which 63.2% Q_{SS} is reached
Definitions		

Symbol	Unit	Description
patent-flow	mL·min ⁻¹	Healthy vascular flow rate threshold of 600
low-flow	mL·min ⁻¹	Representative unhealthy vascular flow rate of 100
no-flow	mL·min ⁻¹	Fully obstructed vascular flow rate of 0

6. There are some grammar errors in the article. For instance, in Fig.3 caption, “FEA simulations (red dashed line) shown”. It is necessary to double check the entire text carefully.

Reviewer #3 (Remarks to the Author):

In this paper, the author describes the flexible and biocompatible thermal sensor for detecting vascular flow variation. The manufacturing process of the sensor was well-described, and in-vitro and in-vivo performance tests were done. I agree with the demand for and importance of this sensor for improving patients' QOL. However, I have several concerns in this paper. I request the author to answer the comments and revise the manuscript.

1. The development and evaluation process was organized very precisely. However, the final output of this paper was only detecting the flow rate variation and was insufficient as a scientific novel knowledge.

Conventional thermal anemometric flow sensing devices are frequently used to measure the velocity of surface fluids in applications such as aircraft and fan blades. In this paper, we aim to measure the flow of blood vessels under the skin without direct contact with the fluid. As a result, the signal strength is weaker. Therefore, it is necessary to focus on theoretical research and explore wearable blood flow sensing. Although simple in application for direct measurements of homogenous materials (e.g. liquids, gasses), non-invasive measurements of heterogeneous materials with underlying fluid flow are much more nuanced. We believe this paper represents one of a very limited number of articles related to wearable thermal anemometry of biofluids:

Our device represents the first example of a wearable anemometer for such measurements of high flow rates (>100 mL/min) and large flow ranges (0-800 mL/min). For example, all previous wearable thermal sensors applied for hydrocephalus⁸⁻¹⁰ (calorimetry), sweat rate^{11,12} (calorimetry), and macrovasculature¹³ (calorimetry) cannot measure these conditions. In consideration of these unique requirements, our device architecture and assembly required a significant redesign within the bounds of a large parameter space (conversion from non-monotonic colorimetric sensing to anemometric sensing, actuator geometry and power density optimization, materials for integration and insulation, and flexible form factor).

This sensing architecture was specifically developed for this clinical application and the downselection/optimization of certain parameters with the use of benchtop data and experimental design resulted in the bespoke design suited for the noninvasive measurement of VA patency. While there are other thermal anemometry-based devices which are used for a number of applications, there are no such devices which are designed and optimized specifically for this application.

Despite the significant need for at-home flow measurements of VAs, there are limited examples of applicable technology that can be used at-home and without skilled operators. We present the first iteration of such a device that has the capacity to be commercialized as a wireless sensor with an integrated mobile device app.

2. All of the figures on temperature variation were described using temperature difference Delta T. The absolute temperature value is also important in evaluating the harmfulness to the tissue.

Therefore, please write the base temperature of Delta T to estimate the absolute value.

For all tissue/material measurements, devices were conformally adhered for ~2 min in the absence of heating in order to measure a baseline temperature ($\Delta T = 0$). Representative baseline tissue surface temperatures were ~21°C for the benchtop model and ~32-37 °C for in vivo measurements. We validate these values with IR (Fig. 2d, Fig 4d,

Supplementary Fig. 8, 9, 19, and 26) and thermocouple measurements (Supplementary Fig. 9, 28, 34). We have included additional data of absolute temperatures of tissue surfaces from Fig. 4c, as measured with a thermocouple, in Supplementary Fig 22. Baseline tissue temperatures are now included in Supplementary Figure captions.

3. The author set the temperature to the operation limit of 52 degrees Celsius.

It should be higher than the denaturation temperature of the protein. The author confirmed the thermal damage in the manuscript; however, the cumulative effect has not been considered. Please explain how the operation temperature was determined in this study.

Responses to these questions are provided in the following text and Figures from the manuscript. Specifically, cumulative dosing and maximal device temperatures are listed in Supplementary Fig. 37:

“FDA safety guidelines allow an increase in local temperatures at the skin-device interface of up to $42 \pm 2 \text{ }^\circ\text{C}$, which corresponds to a $10 \text{ }^\circ\text{C}$ increase over normal skin temperature.³⁵ Our in vivo measurements showed an increase of 3-8 $^\circ\text{C}$ during normal device operation on tissue with underlying blood flow, which does not exceed IEC clinical limit guidelines for devices in contact with tissue.³⁰ Furthermore, histological evaluation of harvested tissues show no remarkable change in the vessel or skin morphology, despite continuous activation (Fig. 7c and Supplementary Figs. 36-37).“

“Subsequent experiments utilize a PD of 1 mW/mm^2 , ensuring maximal heating under no-flow conditions remains below the threshold for high heat pain receptor activation ($\text{TRPV2} > 52 \text{ }^\circ\text{C}$),^{28,29} and within safe limits for clinical dermal devices.³⁰”

4. From the viewpoint of detecting the flow rate variation, what is the minimum heater power to detect the temperature variation? The author developed an accurate and fast response sensor on the device. I think the author can reach the same conclusion even by reducing the heater power.

We show a linear relationship to heater power and temperature under no-flow conditions in Figure 2e-f. In the presence of flow, however, sensitivity (Equation 1 in the manuscript) is impacted by applied power density (PD). For instance, sensitivity is reduced from 33.2% at high PD (2 mW/mm^2) to 23.2% at lower PD (0.55 mW/mm^2). This data (see below) is not included in the manuscript as it was collected on a different vessel material (flexible PVC) than those reported in this manuscript (biomimetic vessel and graft). However, we agree that we have the ability to leverage PD reduction to minimize total thermal dosing and power consumption.

$$\text{Sensitivity (\%)} = (\Delta T_{100} - \Delta T_{800}) / \Delta T_{800} \times 100 \quad (\text{Equation 1})$$

Power Density, PD (mW/mm ²)	Sensitivity (%)	Normalized Sensitivity (%/PD)
2.00	33.2	16.6
1.00	30.5	30.5
0.55	23.2	42.2

5. In an in vivo experiment, how much temperature fluctuates from biological activity?

In addition, what is the minimum heat power to generate a temperature variation more significant than that fluctuation?

During pig measurements, two devices were used simultaneously to study one vessel being manipulated and a control vessel on a neighboring vessel. This included demonstrations of cold/warm saline injections (Figure 5c +

Supplementary Fig. 29a) and vascular stenosis (Supplementary Fig. 32). Furthermore, Supplementary Fig. 19, 20, 22 (new), and 25 show that body movement and device bending do not significantly impact temperature measurements.

- 1 White, F. C., Roth, D. M. & Bloor, C. M. The pig as a model for myocardial ischemia and exercise. *Lab Anim Sci* **36**, 351-356 (1986).
- 2 Sauerbrey, A. *et al.* Establishment of a Swine Model for Validation of Perfusion Measurement by Dynamic Contrast-Enhanced Magnetic Resonance Imaging. *BioMed Research International* **2014**, 390506 (2014). <https://doi.org/10.1155/2014/390506>
- 3 Oglat, A. A. *et al.* A Review of Medical Doppler Ultrasonography of Blood Flow in General and Especially in Common Carotid Artery. *J Med Ultrasound* **26**, 3-13 (2018). https://doi.org/10.4103/jmu.Jmu_11_17
- 4 Wang, F. *et al.* Flexible Doppler ultrasound device for the monitoring of blood flow velocity. *Science Advances* **7**, eabi9283 (2021). <https://doi.org/doi:10.1126/sciadv.abi9283>
- 5 Monnet, X. & Teboul, J.-L. Transpulmonary thermodilution: advantages and limits. *Critical Care* **21**, 147 (2017). <https://doi.org/10.1186/s13054-017-1739-5>
- 6 Franklin, D. *et al.* Synchronized wearables for the detection of haemodynamic states via electrocardiography and multispectral photoplethysmography. *Nature Biomedical Engineering* **7**, 1229-1241 (2023). <https://doi.org/10.1038/s41551-023-01098-y>
- 7 Boutry, C. M. *et al.* Biodegradable and flexible arterial-pulse sensor for the wireless monitoring of blood flow. *Nature Biomedical Engineering* **3**, 47-57 (2019). <https://doi.org/10.1038/s41551-018-0336-5>
- 8 Krishnan, S. R. *et al.* Continuous, noninvasive wireless monitoring of flow of cerebrospinal fluid through shunts in patients with hydrocephalus. *NPJ Digit Med* **3**, 29 (2020). <https://doi.org/10.1038/s41746-020-0239-1>
- 9 Rajasekaran, S., Qu, H. & Zakalik, K. in *2015 IEEE SENSORS*. 1-4.
- 10 Krishnan, S. R. *et al.* Epidermal electronics for noninvasive, wireless, quantitative assessment of ventricular shunt function in patients with hydrocephalus. *Science Translational Medicine* **10**, eaat8437 (2018). <https://doi.org/doi:10.1126/scitranslmed.aat8437>
- 11 Kwon, K. *et al.* An on-skin platform for wireless monitoring of flow rate, cumulative loss and temperature of sweat in real time. *Nature Electronics* **4**, 302-312 (2021). <https://doi.org/10.1038/s41928-021-00556-2>
- 12 Brueck, A., Iftexhar, T., Stannard, A. B., Yelamarthi, K. & Kaya, T. A Real-Time Wireless Sweat Rate Measurement System for Physical Activity Monitoring. *Sensors* **18**, 533 (2018).
- 13 Webb, R. C. *et al.* Epidermal devices for noninvasive, precise, and continuous mapping of macrovascular and microvascular blood flow. *Sci Adv* **1**, e1500701 (2015). <https://doi.org/10.1126/sciadv.1500701>

REVIEWER COMMENTS

Reviewer #1 (Remarks to the Author):

In this revision, the authors have included additional details that clarify the design of the sensor and the experiments. I have additional questions arising from the authors' response to the reviewers' comments.

1. There is a disconnect between what is claimed in the title/abstract and the reported sensor. As all reviewers have pointed out, the sensor's level of integration is below what is expected of paper published in this journal. While the authors make the point that the main advance is the optimized configuration of the heaters/sensors (and not the integrated system), this is not clear from the title, abstract, and introduction. In my view, the integration of the sensor into a wireless device, while perhaps tedious, is of significant scientific value because it enables testing in continuous monitoring scenarios where a tethered setup cannot be used.

We thank the reviewer for this valid statement. We have updated the title to: "Wearable *wired* thermal sensor for the continuous assessment of flow in vascular access". The abstract now emphasizes that wireless adaptation is needed prior to practical application for CKD patients.

2. Based on the authors' response to my Q1, the current results do not demonstrate in vivo monitoring for BOTH the flow rate and vessel depth expected during vascular access monitoring. It remains unclear if the results from the porcine experiments are applicable when the sensor is separated by heterogenous tissue layers. Additional data should be included to support the effectiveness of the sensor, such as by placing artificial skin over the porcine vessel as suggested by the authors.

We appreciate the reviewer's detailed feedback, which has significantly contributed to enhancing the rigor and applicability of our study. We would like to highlight that Fig. 4c illustrates the difference in ΔT_{Heater} between measurements taken from the chest (no underlying flow) and the neck (with underlying flow), as described in the Methods Section on "Transdermal Assessment of Vascular Flow." Despite the substantial tissue depth of ~20 mm (now updated in the text, referring to the y-axis of **Supplementary Fig. 22c**) for the carotid artery beneath heterogeneous tissue layers (epidermis, dermis, and subcutaneous fat and a muscle layer), our sensor successfully detected flow-induced differences. This demonstrates the sensor's effectiveness even under these challenging conditions, where heterogeneous tissue and substantial tissue depth could potentially impede accurate flow detection. This is now discussed in the manuscript text.

To further address the reviewer's concerns regarding the applicability of our sensor through heterogeneous tissue layers, we conducted additional experiments using a biomimetic vessel pad (SynDaver, SKU:160553). This model includes artificial arteries embedded 3 mm beneath an epidermal/dermal/subcutaneous fat tissue layer that closely mimics the geometry, thermal conductivity, and mechanical properties of human tissue, and is consistent with the biomimetic vessel model described in our manuscript. The tissue model simulates a depth of 3 mm of primarily comprised of subcutaneous fat (low thermal conductivity), thereby providing a realistic scenario for fistula measurements over a thicker than average skin layer (see **Supplementary Table 5**). Our experiments revealed that sensor sensitivity decreases from 24.21% at a tissue depth of 1.89 mm to 5.59% at a depth of 3.00 mm. These findings have been incorporated into our manuscript and are now included as **Supplementary Fig. 17**. Similar commercially available models at different tissue depths were not accessible.

Our in vivo and benchtop results clearly demonstrate the sensor's capability to detect vascular flow at higher depths and with heterogenous overlying tissues. For practical applications, it is essential to calibrate the devices to each patient's specific skin thickness to ensure accurate and individualized monitoring of vascular access. This calibration

can be effectively performed using methods similar to those described by Madhvapathy et al. (referenced in our manuscript), thereby optimizing the sensor's performance for each unique clinical scenario.

Reviewer #2 (Remarks to the Author):

In this paper, a soft, wearable, flexible device was proposed to continuously monitor blood flow changes. The authors used in-vitro and in-vivo experiments to demonstrate that the sensor can sensitively detect vascular access failure during hemodialysis. It can be seen that the sensor has quite high sensitivity through the results presented in the paper. Although the sensor has the potential to be used in medical field in the future, there are still several unavoidable questions about the current work.

Some main issues are listed as follows:

1. In the paper, the authors have optimized the shape of the sensor, but whether the narrow and long sensor depends more on the precise alignment of blood vessels to ensure the accuracy of the measurement?

We agree with the reviewer that device misalignment can affect flow accuracy. In this manuscript, we discuss the advantage of the anemometry principle in its robustness to mounting errors. In traditional calorimetric blood flow devices¹⁻³, the temperature of the upstream and downstream center points are used as the characteristic signal. This results in a harmful effect on the temperature signal once the mounting deviates from the top of the blood vessel. In contrast, this device uses the average temperature of the entire heating area as the characteristic signal, which is 450 mm², which reduces sensitivity to mounting errors. **Supplementary Fig. 5d** highlights the higher tolerance of our large heater over that of a small sensor when varying contact. In addition, compared with most blood vessels that are buried deep in the skin and cannot be observed with the naked eye, fistulas and grafts are much larger (about 10 times the inner diameter) and usually present as superficial blood vessels protruding from the skin, which is easy to observe, making mounting errors smaller.

2. The author described the experimental content in great detail, in contrast, the analysis of the measurement principle and theoretical model is rough.

According to the suggestion, we now describe our measurement principle in more detail within the manuscript text (see tracked changes), and have significantly extended the methods section on FEA experiments performed in **Figures 3-4**. Furthermore, we now include **Equation 2**, the extended Steinhart-Hart equation, which relates measured resistances with calculated device temperatures. Representative MATLAB scripts and raw data from LabView measurements are now included as part of the Code Availability section (**Code Ocean DOI:10.24433/CO.5920840.v1**), which details our complete data processing and signal analysis procedure. Source data and custom LabView data acquisition files are provided as **Supplementary Datasets**.

3. In the in-vitro experiment, the author used only one thickness of skin to attach to the blood vessels, and the experiment was inadequate and unconvincing.

We appreciate the reviewer's detailed feedback, which has significantly contributed to enhancing the rigor and applicability of our study. We would like to highlight that Fig. 4c illustrates the difference in ΔT_{Heater} between measurements taken from the chest (no underlying flow) and the neck (with underlying flow), as described in the Methods Section on "Transdermal Assessment of Vascular Flow." Despite the substantial tissue depth of ~20 mm (now updated in **Supplementary Fig. 22**) for the carotid artery beneath heterogeneous tissue layers (epidermis, dermis, and subcutaneous fat and a muscle layer), our sensor successfully detected flow-induced differences. This demonstrates the sensor's effectiveness even under these challenging conditions, where heterogeneous tissue and substantial tissue depth could potentially impede accurate flow detection. This is now discussed in the manuscript text.

To further address the reviewer's concerns regarding the applicability of our sensor through heterogeneous tissue layers, we conducted additional experiments using a biomimetic vessel pad (SynDaver, SKU:160553). This model includes artificial arteries embedded 3 mm beneath an epidermal/dermal/subcutaneous fat tissue layer that closely mimics the geometry, thermal conductivity, and mechanical properties of human tissue, and is consistent with the biomimetic vessel model described in our manuscript. The tissue model simulates a depth of 3 mm of primarily comprised of subcutaneous fat (low thermal conductivity), thereby providing a realistic scenario for fistula measurements over a thicker than average skin layer (see **Supplementary Table 5**). Our experiments revealed that sensor sensitivity decreases from 24.21% at a tissue depth of 1.89 mm to 5.59% at a depth of 3.00 mm. These findings have been incorporated into our manuscript and are now included as **Supplementary Fig. 17**. Similar commercially available models at different tissue depths were not accessible. Our in vivo and benchtop results clearly demonstrate the sensor's capability to detect vascular flow at higher depths and with heterogenous overlying tissues.

4. Does the sensor need to be recalibrated each time the test object is changed?

For practical applications, it is essential to calibrate the devices to each patient's specific tissue depth and type (fistula vs graft) to ensure accurate and individualized monitoring of vascular access. This calibration can be effectively performed using methods similar to those described by Madhvapathy et al. (referenced in our manuscript), thereby optimizing the sensor's performance for each unique clinical scenario.

5. In view of individual differences, whether differences in the skin thermal conductivity above the blood vessels affect the measurement results?

The epidermis (~100 μm in humans) is subject to dehydration and changes in net skin thermal conductivity. However, the underlying dermis (~1-4 mm) which directly surround blood vessels exhibit little dependence upon hydration conditions⁴. Reports by Madhvapathy, et. al⁵ demonstrate that small heaters (0.9x0.9 mm²) actuated for short times (2 s) at a PD = 10 mW/mm² induces sufficient thermal penetration (*total power = 8.1mW*) to a depth of 1

mm for accurate hydration (and subsequently thermal conductivity) measurements. They note that thermal penetration depth increases with heater size and power, which our device exceeds ($60 \times 7.5 \text{ mm}^2$ heater, $\text{PD} = 1 \text{ mW/mm}^2$) with a *total power = 450 mW*. Furthermore, when they increase the measurement time from 2 s to 13 s, their thermal penetration depths increase to 6 mm. Given the long-term measurements with our devices (400s), physiologically relevant thermal conductivity differences in the top layer of our tissues will have a negligible effect on our overall flow readings. This is further emphasized by skin dehydration experiments in **Supplementary Fig. 17**. We now discuss this in the manuscript text.

6. The network of blood vessels in the human body is very complex, whether the sensor has the same high sensitivity for tilted vessels with varying depth?

This is a valid consideration. We anticipate that varying tissue depths across the device length will have an impact on readings. As such, an effective device that is minimally influenced by variable tissue depths across the length of the device will rely on a shorter form factor, which is outside the scope of this study. Our conclusions section discusses the need for further miniaturization and now emphasizes this need in the context of variable tissue depths.

Reviewer #3 (Remarks to the Author):

The author responded appropriately to the reviewer's comments.

Reviewer #3 (Remarks on code availability):

N/A

- 1 Krishnan, S. R. *et al.* Continuous, noninvasive wireless monitoring of flow of cerebrospinal fluid through shunts in patients with hydrocephalus. *npj Digital Medicine* **3**, 29 (2020). <https://doi.org:10.1038/s41746-020-0239-1>
- 2 Krishnan, S. R. *et al.* Epidermal electronics for noninvasive, wireless, quantitative assessment of ventricular shunt function in patients with hydrocephalus. *Science Translational Medicine* **10** (2018).
- 3 Webb, R. C. *et al.* Epidermal devices for noninvasive, precise, and continuous mapping of macrovascular and microvascular blood flow. *Sci Adv* **1**, e1500701 (2015). <https://doi.org:10.1126/sciadv.1500701>
- 4 Madhvapathy, S. R. *et al.* Advanced thermal sensing techniques for characterizing the physical properties of skin. *Applied Physics Reviews* **9** (2022). <https://doi.org:10.1063/5.0095157>
- 5 Madhvapathy, S. R. *et al.* Reliable, low-cost, fully integrated hydration sensors for monitoring and diagnosis of inflammatory skin diseases in any environment. *Science Advances* **6**, eabd7146 (2020). <https://doi.org:doi:10.1126/sciadv.abd7146>

REVIEWER COMMENTS

Reviewer #1 (Remarks to the Author):

I thank the authors for their effort to address the questions raised. All of my concerns have been satisfactorily addressed. Regarding the title, I personally feel that "A soft thermal sensor for the continuous assessment of flow during vascular access" is both accurate and attractive. This avoids describing the sensor as both "wearable" and "wired", which is somewhat contradictory. I leave the final form at the discretion of the authors and editor.

We agree with the reviewer that this is a more appropriate title. It has been amended.

Reviewer #2 (Remarks to the Author):

In this paper, a soft, wearable, flexible device was proposed to continuously monitor blood flow changes. It can be seen that the authors did a lot of experiments to verify the performance of the sensor. The authors have explained some of the possible problems in the response letter. Beyond that, here are still some additional questions about the current work.

Some main issues are listed as follows:

1. Given that we have done similar experiments before, it is generally easy to distinguish between fluid flow and unflow, but it is difficult to distinguish between fluids with varying flow rates. As can be seen in the article, the difference in temperature rise curves for different flow rates is very small. In fact, since the difference is so small that any experimental error may affect the sensitivity of the results, how can the authors accurately distinguish the temperature rise curves for different flow rates?

We thank the referee for this important input. We agree that the changes in the temperature rise curves can be small for different flow rates in the regime of high flow rates. From a practical standpoint, experimental error due to environmental fluctuations or other forms of noise could be significant in such cases. To address this issue, our current work focuses on exploiting co-integrated, adjacent flow sensors, to allow for a differential measurement approach, with enhanced robustness to noise. We added the following statement to a new "Limitations" section – "As indicated by experimental measurements and simulation results, the temperature curves depend relatively weakly on flow rate in the regime of high flow rate. Thus, from a practical standpoint, experimental error due to environmental fluctuations or other forms of noise could be significant in such cases. One future approach to explore exploits co-integrated, adjacent flow sensors, to allow for differential measurements, with decreased sensitivity to noise."

2. When the heating power is $1\text{mW}/\text{mm}^2$, the temperature rise is very close to that of the pain receptors, and while considering the measurement accuracy, the authors should consider reducing the heating power appropriately to alleviate the discomfort that may exist when wearing the product.

We agree that heating powers should be maintained well below the thresholds for activating pain receptors. Ongoing work explores the use of cooling, rather than heating, as the basis for measurement. The thresholds for activating pain receptors are larger for cooling than for heating. We added the following statement in a new "Limitations" section – "The measurement accuracy can be improved by use of cooling rather than heating, simply because the threshold changes in temperature for activating pain receptors are larger for cooling."

3. When wearing the measurement for a long period of time, is there any heat storage in the structure of the sensor itself, causing the sensitivity or accuracy of the measurement to be affected?

We thank the referee for this comment. We agree that measurements for long periods of time could lead to adverse effects of cumulative heating. Because rapid changes in flow are not expected, in practice, the measurements will be performed in a low duty cycle mode, perhaps once every one or two hours. In this way, cumulative heating can be neglected. We added the following statement in a new "Limitations" section – "Another consideration in practical use is that continuous measurements for long periods of time could lead to adverse effects of cumulative

heating. Because rapid changes in flow are not expected, in practice, the measurements will be performed in a low duty cycle mode, perhaps once every one or two hours. In this way, cumulative heating can be neglected.”